# Towards Trustworthy Video Anomaly Understanding: A Class-Guided Chain-of-Evaluation Metric and An Anomaly-focused Meta-Benchmark

**Jiaxu Leng** [1 2]  **Zhoujie Huang** [1 2]  **Mingpi Tan** [1 2]  **Zhanjie Wu** [1 2]  **Xinbo Gao** [1 2]

## Abstract

The trustworthiness of evaluation is critical to reliable model comparison and deployment in Video Anomaly Understanding (VAU). However, existing metrics are sensitive to expression styles and normal content, and this field lacks a diagnostic benchmark to validate metric validity and robustness. To bridge this gap, we propose: (1) a Class-Guided Chain-of-Evaluation (CG-CoE) metric, which structures assessment by extracting anomalous events and matching them under a class-specific semantic tolerance boundary, thereby decoupling anomaly semantics from expression styles; and (2) an anomaly-focused meta-evaluation benchmark with two subsets: Anomalous Event-level Annotations (AEA) subset for measuring the validity of reflecting VAU models' anomaly understanding ability and Controlled Variant Pairs (CVP) with fixed anomalies for quantifying robustness to stylistic perturbations. Extensive experiments demonstrate that CG-CoE achieves state-of-the-art (SOTA) validity and robustness.

## 1. Introduction

Unlike Video Anomaly Detection (VAD) (Huang et al., 2025b; Li et al., 2025a; Leng et al., 2024), VAU (Zhu et al., 2025a; Huang et al., 2025a) aims not only to identify but also to describe and analyze video anomalies. Instead of merely producing a sequence of frame-level anomaly scores, VAU addresses richer questions, including what anomalous events occur, where they take place, how they unfold, and why they happen. This shift is motivated by practi-

cal needs in surveillance and other safety-critical settings, where an alarm must accurately describe the anomaly to support downstream decisions.

To meet these requirements, most existing VAU methods employ Multimodal Large Language Models (MLLMs) to endow the system with strong understanding capabilities. However, despite improving VAU models via both training-free approaches (Ye et al., 2025; Chen et al., 2025) and training-based approaches (Zhang et al., 2025; Du et al., 2024b; Tang et al., 2024), prior work largely overlooks a key obstacle to real-world deployment: the lack of principled evaluation metrics and, crucially, meta-evaluation benchmarks to validate metrics themselves.

When evaluating VAU model responses, a metric should primarily reflect whether the anomalous events in the prediction align with the label, rather than overall similarity. As shown at the top of Fig. 1, two correct predictions can either paraphrase the same anomaly in a different expression style (Pred1) or mention the anomaly after a long normal context (Pred2), yet holistic similarity metrics assign low scores in both cases. This failure stems from holistic matching: N-gram-based metrics (Papineni et al., 2002; Lin, 2004) rely on lexical overlap, while Embedding-based metrics (Zhang et al., 2019; Zhao et al., 2019) and LLM-based metrics (Tang et al., 2024; Huang et al., 2025a; Zhu et al., 2025a) aggregate normal and anomalous content, causing scores to drift with style and be dominated by abundant normal details. **Therefore, there is an urgent need for an anomaly-focused metric that reflects the alignment of anomalous events and is robust to diverse expression styles.**

Moreover, VAU lacks a meta-evaluation benchmark to assess metrics themselves, especially whether they truly focus on anomalous events. By contrast, summarization (Honovich et al., 2022; Zhao et al., 2023), open-domain question answering (Roh et al., 2025; Yue et al., 2025), and machine translation (Deutsch et al., 2023b;a) have established meta-evaluation benchmarks that quantify agreement with human judgments, robustness to controlled perturbations, and related aspects, thereby validating metrics before they are used to assess models. Without such a benchmark, reported improvements may reflect artifacts of unreliable metrics

[1]School of Computer Science and Technology, Chongqing University of Posts and Telecommunications, Chongqing, China [2]Chongqing Institute for Brain and Intelligence, Guangyang Bay Laboratory, Chongqing, China. Correspondence to: Xinbo Gao <gaoxb@cqupt.edu.cn>.

*Proceedings of the 43rd International Conference on Machine Learning*, Seoul, South Korea. PMLR 306, 2026. Copyright 2026 by the author(s).

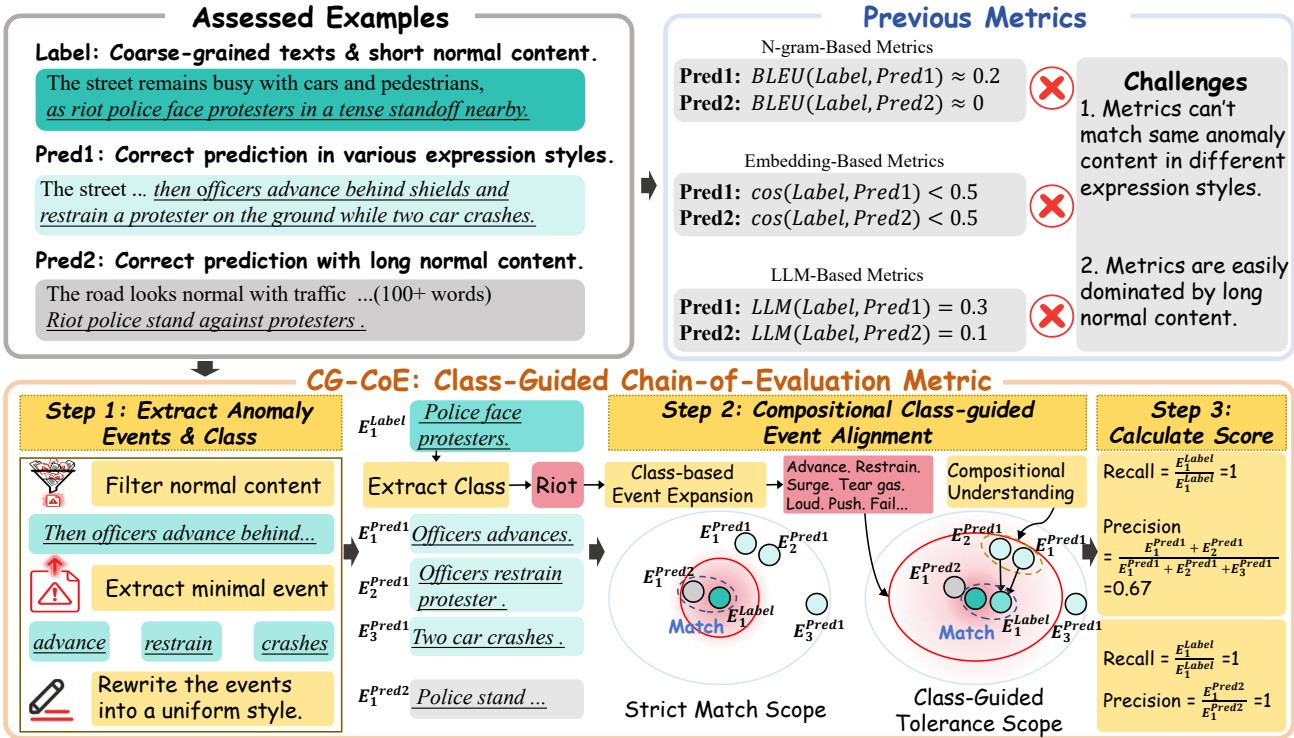

*Figure 1.* **Overview of CG-CoE.** Previous metrics drifts with expression style and long normal content. CG-CoE instead extracts minimal anomalous events and class, aligns them via class-guided compositional matching, and computes event-level Precision/Recall for robust evaluation.

rather than genuine progress. **This gap reveals another urgent need for a dedicated benchmark to evaluate the anomaly-focusedness of VAU metrics.**

To satisfy the first need, we propose CG-CoE, a Class-Guided Chain-of-Evaluation metric for anomaly-focused assessment (Fig. 1 (bottom)), inspired by the human annotation process. During annotation, humans leverage class-level common knowledge to decide which discrepancies are tolerable under the anomaly class. Accordingly, in Step 1, CG-CoE extracts minimal anomalous events from both the label and the prediction and rewrites them into a uniform style, thereby suppressing normal content and expression styles; it also extracts the anomaly class from the label as a hint. In Step 2, instead of strict semantic matching, CG-CoE matches events within a class-defined tolerance boundary, allowing differences in granularity and detail as long as they remain plausible within the corresponding class. Finally, CG-CoE computes event-level Precision and Recall from the matches, providing a decoupled view of anomaly understanding: Precision measures the proportion of predicted anomalies that supports the label, whereas Recall measures how completely they cover the labeled anomalies.

To satisfy the second need, we introduce an anomaly-focused meta-evaluation benchmark with two subsets: AEA for validity in reflecting VAU models' anomaly understand-

ing, and CVP for robustness to diverse expression styles (Fig. 2). In AEA, annotators extract and match anomalous events from labels and predictions to produce event-level Precision/Recall as human judgments of anomaly understanding (Fig. 3). A metric is more valid if its scores better agree with these human rankings, measured by Spearman's rank correlation $\rho$ (higher is better). In CVP, we fix an anomaly anchor, which preserves anomaly semantics, and generate paired variants while perturbing a sampled expression-style constraint from a predefined taxonomy. Robustness is quantified by the mean within-group score variation $\Delta$ (lower is better).

In summary, our contributions are threefold: (1) We propose CG-CoE, a VAU-specific metric that evaluates anomaly understanding by extracting anomalous events and performing evidence-backed matching under a class-guided tolerance boundary. (2) We introduce the first anomaly-focused meta-evaluation benchmark for VAU, comprising AEA for validity via human event-level Precision/Recall annotations and CVP for robustness via controlled variants with fixed anomaly semantics. (3) Using this benchmark, we show that CG-CoE achieves SOTA validity and robustness, providing a practical basis for trustworthy VAU evaluation and model deployment.

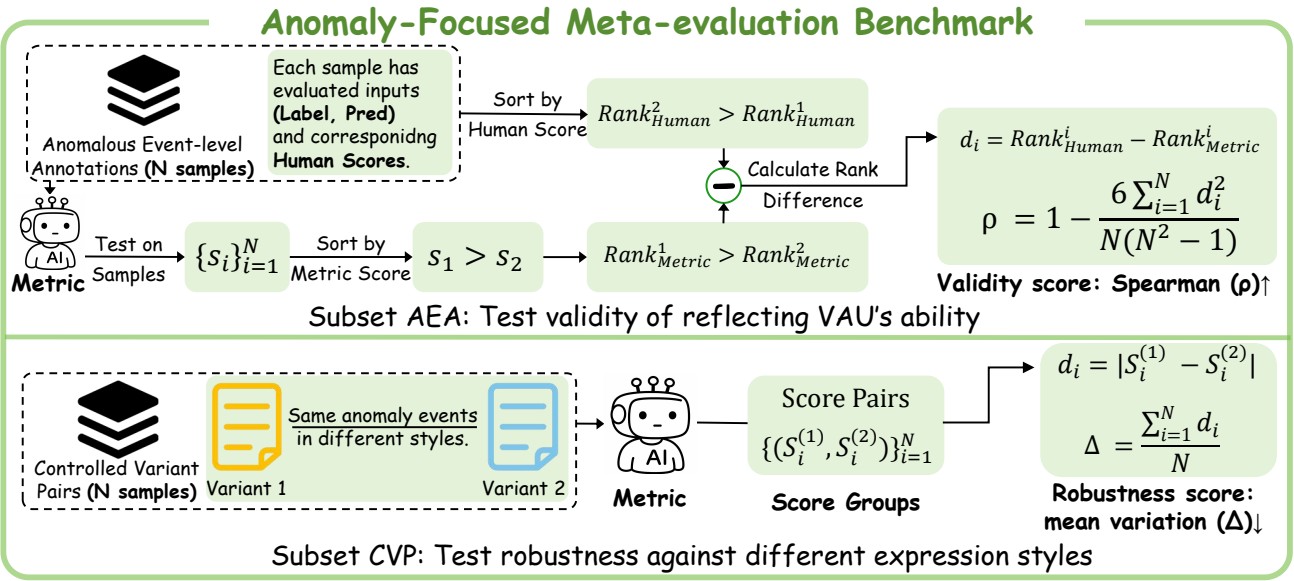

*Figure 2.* **Anomaly-focused meta-evaluation benchmark.** Top: Human scores come from our human-annotated event-level Precision and Recall. During evaluation, we compute Spearman's rank correlation $\rho$ (higher is better) between the metric scores and each human score separately. Bottom: CVP measures robustness by the mean score variation $\Delta$ (lower is better) over controlled stylistic variant pairs.

**Conflict of Interest Disclosure.** The authors declare no financial conflicts of interest related to this work.

## 2. Related Work

### 2.1. From Video Anomaly Detection to Understanding

VAD aims to identify anomalous events in videos. Most traditional VAD methods (Hu et al., 2025; Amicantonio et al., 2025; Leng et al., 2022; 2025b; Majhi et al., 2025; Leng et al., 2025a; Li et al., 2025c; Mo et al., 2026; Li et al., 2025b) output only an anomaly score or a discrete class label, making it difficult to understand which aspects of an abnormal event the model captures or overlooks. To obtain more interpretable results, several recent studies (Zhang et al., 2024; Ye et al., 2025; Du et al., 2024a; Feuerstein et al., 2023; Ma et al., 2025) leverage MLLMs to analyze, describe, and explain anomalies beyond detection. For example, HAWK (Tang et al., 2024) formulates open-world VAU and integrates motion information into MLLMs to enhance anomaly understanding. Holmes-VAU (Zhang et al., 2025) further extends this paradigm to multi-granularity VAU and designs an anomaly-focused temporal sampler that better captures anomalies in long videos.

### 2.2. Metrics in VAU

Despite the rapid advancements in Video Anomaly Understanding (VAU), existing metrics often fail to focus on anomalies. They can be grouped into three families. First, N-gram-based metrics (Lin, 2004; Vedantam et al., 2015), such as BLEU (Papineni et al., 2002), predominantly measure lexical overlap and are insensitive to semantic equivalence. Second, model-based metrics built on pretrained language models (Yuan et al., 2021), such as BERTScore (Zhang et al., 2019) and UniEval (Zhong et al., 2022), better capture semantic similarity, but they still assess overall similarity and can be dominated by normal content. Most recently, LLM-based metrics (Liu et al., 2023; Zhu et al., 2025b) have emerged. Some VAU works, such as VAU-R1 (Zhu et al., 2025a) and HAWK (Tang et al., 2024), rely on prompt-based LLM evaluation, instructing the LLM to focus its judgment on anomalous events when scoring. However, these metrics can still drift with normal content and expression style.

### 2.3. Meta-Evaluation Benchmarks

Meta-evaluation benchmarks serve as the standard for validating the quality of metrics. Extensive benchmarks (Wang et al., 2025; Deutsch et al., 2023a; Yue et al., 2025; Honovich et al., 2022; Wang et al., 2023; Li et al., 2023) have been developed for Natural Language Generation (NLG) tasks. In summarization, SummEval (Fabbri et al., 2021) revealed the fragility of human–metric correlations. In machine translation, the WMT Metrics Shared Task (Freitag et al., 2024) has become a widely used benchmark for monitoring metric reliability and stability. Despite these advancements, the field of Video Anomaly Understanding lacks a meta-evaluation framework, leaving a gap for real-world applications.

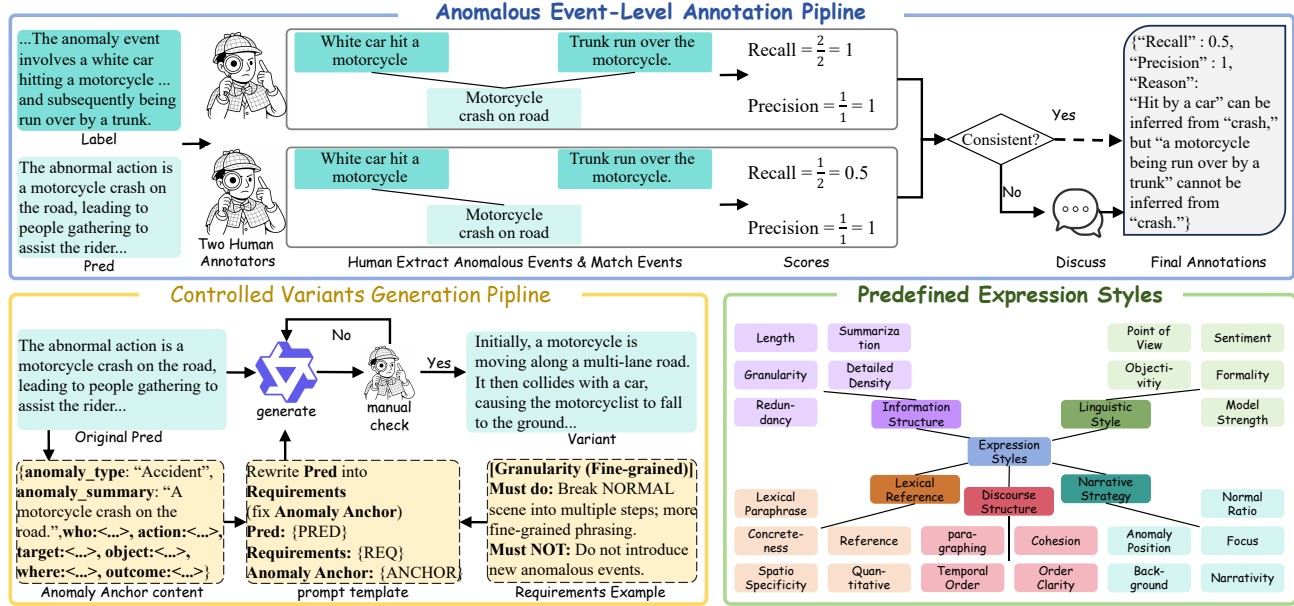

*Figure 3.* **Benchmark construction pipeline and Predefined expression styles.** Top: two-annotator event extraction, matching, and adjudication yields event-level Precision/Recall. Bottom left: CVP fixes anomaly anchors and perturbs predefined expression style dimensions to generate validated variants. Bottom right: predefined expression style dimensions.

## 3. Meta-Evaluation Benchmark

### 3.1. Overview and Notation

In VAU, each instance consists of a video, a question, and a reference label text $l$. Given the video and question, a VAU model outputs a textual prediction $\hat{a}$. A metric (evaluator) $E$ assigns a scalar score:

$$s = E(\hat{a}, l), \quad s \in \mathbb{R}. \tag{1}$$

We evaluate anomaly-focusedness via AEA (Sec. 3.2) and CVP (Sec. 3.3) .

### 3.2. Subset: Anomalous Event-level Annotations

AEA provides human event-level Precision ($\mathrm{Prec}^h$) and Recall ($\mathrm{Rec}^h$) to measure anomalous-event alignment: $\mathrm{Prec}^h$ reflects hallucinated anomalous events, while $\mathrm{Rec}^h$ reflects missing anomalous events. It contains $N$ $(\hat{a}, l)$ pairs:

$$\mathcal{D}_{\mathrm{AEA}} = \left\{ \left( \hat{a}_i, l_i, \mathrm{Prec}_i^h, \mathrm{Rec}_i^h \right) \right\}_{i=1}^{N}$$

To evaluate $E$, we obtain its scores $\{s_i\}_{i=1}^{N}$ on $\mathcal{D}_{\mathrm{AEA}}$ using Eq. 1 and measure validity by Spearman's rank correlation:

$$\rho_{\mathrm{Prec}} = \mathrm{Spearman}\left( \{s_i\}_{i=1}^{N}, \{\mathrm{Prec}_i^h\}_{i=1}^{N} \right). \tag{2}$$

$$\rho_{\mathrm{Rec}} = \mathrm{Spearman}\left( \{s_i\}_{i=1}^{N}, \{\mathrm{Rec}_i^h\}_{i=1}^{N} \right). \tag{3}$$

Higher $\rho_{\mathrm{Prec}}$ indicates that $E$ more reliably reflects hallucinated anomalous events, while higher $\rho_{\mathrm{Rec}}$ indicates that $E$ more reliably reflects missing anomalous events.

| Dataset | Human meta-annotations | VAU Field | Event-level annotations | Test robustness to styles |
|---|---|---|---|---|
| VAU datasets | | | | |
| HAWK | ✗ | ✓ | ✗ | ✗ |
| CUVA | ✗ | ✓ | ✗ | ✗ |
| Meta-benchmark datasets | | | | |
| ScholarSum | ✓ | ✗ | ✗ | ✗ |
| EVOUNA | ✓ | ✗ | ✗ | ✗ |
| Ours (AEA+CVP) | ✓ | ✓ | ✓ | ✓ |

*Table 1.* **Dataset comparison.** Our proposed benchmark is the first meta-evaluation benchmark for VAU, providing anomalous event-level annotations and controlled variant pairs.

#### 3.2.1. ANNOTATION PROTOCOL

All AEA annotations were produced by two domain experts in VAU, with substantial expertise in identifying and interpreting anomalous events, enabling reliable event-level annotation. Annotators compute $(\mathrm{Prec}^h, \mathrm{Rec}^h)$ using an extract-and-match protocol.

**Step 1: extract anomalous events.** From $l$ and $\hat{a}$, annotators extract anomalous-event sets $\mathcal{E}^l$ and $\mathcal{E}^{\hat{a}}$. An event is recorded only when its associated evidence span explicitly supports both (i) the event content and (ii) the assigned anomaly class. If the text merely states a class label without describing the corresponding event, it is not counted as an event. Within each text, multiple paraphrases that clearly describe the same underlying event are merged into a single event to avoid double counting.

*Table 2.* **Inter-annotator agreement before adjudication.**

|         | Precision | Recall |
|---------|-----------|--------|
| ICC(2,1)| 0.85      | 0.82   |

**Step 2: match events by incident meaning and evidence.**
Matches are represented as a bipartite relation $M^h \subseteq \mathcal{E}^{\hat{a}} \times \mathcal{E}^l$. A pair $(e, e') \in M^h$ is added when $e$ and $e'$ describe the same event, requiring overlapping incident evidence. Class agreement without aligned evidence does not constitute a match.

**Step 3: compute human precision and recall.** Let $n_{\hat{a}} = |\mathcal{E}^{\hat{a}}|$ and $n_l = |\mathcal{E}^l|$. Let $n_{\hat{a}}^m$ (resp., $n_l^m$) denote the number of prediction-side (resp., label-side) events that have at least one match under $M^h$. When $n_{\hat{a}} > 0$ and $n_l > 0$,

$$\text{Prec}^h = \frac{n_{\hat{a}}^m}{n_{\hat{a}}}, \qquad \text{Rec}^h = \frac{n_l^m}{n_l}. \qquad (4)$$

If $\mathcal{E}^{\hat{a}} = \mathcal{E}^l = \emptyset$, we set $(\text{Prec}^h, \text{Rec}^h) = (1, 1)$; otherwise, if exactly one side is empty, we set $(\text{Prec}^h, \text{Rec}^h) = (0, 0)$.

### 3.2.2. DATA SOURCE AND STATISTICS

AEA is derived from the Holmes-VAU (Zhang et al., 2025) test split. For each video, we construct $l$ by integrating annotations from both the description and classification tasks. Using a unified prompt across representative VAU models, we generate $\hat{a}$ and obtain 900 $(\hat{a}, l)$ pairs from 150 videos. Two trained annotators independently label all pairs and resolve disagreements through discussion to produce consensus annotations. We remove a small number of irrelevant or non-responsive predictions, yielding $N = 865$ pairs.

We report inter-annotator agreement before discussion in Table 2, suggesting that the guidelines are clear and the human annotations are reliable and reproducible, which strengthens the credibility of AEA. We report the distribution of $(\text{Prec}^h, \text{Rec}^h)$ in Fig. 4. The concentration at $(0, 0)$ and $(1, 1)$ is mainly due to (i) relatively few multi-anomaly cases in our current test split and (ii) conservative model predictions that focus on the primary anomaly. Thus, predictions are often either fully correct or miss the key anomaly, and the resulting extremes suggest our event-level annotations are driven by anomaly evidence rather than irrelevant content. The full annotation protocol, failure cases, and implementation details are provided in Appendix B.

### 3.3. Subset: Controlled Variant Pairs

CVP evaluates robustness by measuring score consistency under anomaly-invariant rewrites (surface-form changes only). CVP contains $G$ groups: $\mathcal{D}_{\text{CVP}} = \left\{ \left( \hat{a}_g^{(1)}, \hat{a}_g^{(2)}, l_g \right) \right\}_{g=1}^G$.

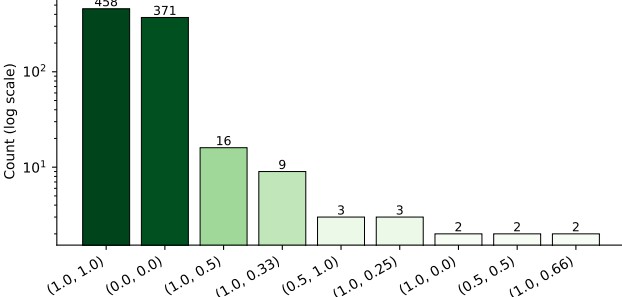

*Figure 4.* **Distribution of** $(\text{Prec}^h, \text{Rec}^h)$.

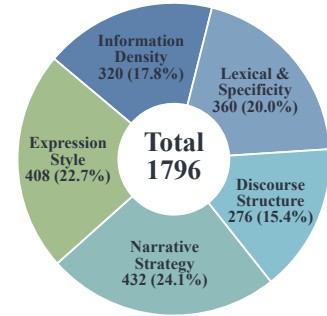

*Figure 5.* **Distribution of randomly selected dimensions.**

where $\hat{a}_g^{(1)}$ and $\hat{a}_g^{(2)}$ are two variants that preserve the same anomaly content with respect to $l_g$.

To evaluate robustness of $E$ on $\mathcal{D}_{\text{CVP}}$, we compute its scores: $s_g^{(1)} = E\left( \hat{a}_g^{(1)}, l_g \right), s_g^{(2)} = E\left( \hat{a}_g^{(2)}, l_g \right)$, and define the within-group score variation

$$\Delta_g = \left| s_g^{(1)} - s_g^{(2)} \right|. \qquad (5)$$

We aggregate over all groups:

$$\mathbb{E}[\Delta] = \frac{1}{G} \sum_{g=1}^G \Delta_g, \qquad (6)$$

where lower $\mathbb{E}[\Delta]$ indicates that $E$ is more invariant to expression styles that preserve anomaly semantics.

### 3.3.1. CONSTRUCTION PRINCIPLE

CVP enforces anomaly invariance via extracting explicit anomaly anchors and introduces controlled surface differences using a predefined edit taxonomy.

**Anomaly anchors.** For each $(l, \hat{a})$, we extract an anomaly anchor from $\hat{a}$ that specifies the anomaly semantics to preserve, including anomaly classes, a brief summary, and a structured list of anomalous events (e.g., who, action, object_or_target, where_when, outcome). Anchors are extracted with a dedicated prompt and serve as the sole reference during generation and verification to check

whether a rewrite introduces new anomalies, removes anchored anomalies, or changes key outcomes.

**Controlled edit taxonomy.** We define a two-level taxonomy of anomaly-irrelevant transformations with 5 major dimensions and 24 sub-dimensions. For each sub-dimension, we provide an explicit definition and operational guidelines that specify what to edit and what to keep unchanged, preventing unintended changes to anomaly semantics. For each $\hat{a}$, we sample exactly one sub-dimension as the only editing constraint, enabling robustness diagnosis by dimension.

### 3.3.2. GENERATION AND QUALITY CONTROL

**Source data and variant generation.** CVP is constructed from the same Holmes-VAU-derived $(\hat{a}, l)$ pool used for AEA, starting from 900 $(l, \hat{a})$ pairs. For each $(l, \hat{a})$, we first extract an anomaly anchor from $\hat{a}$. We then perform two independent rounds of variant generation: in each round, we randomly sample one sub-dimension from the edit taxonomy and generate one variant with Qwen3-32B (Yang et al., 2025), conditioned on the anomaly anchor and the sampled sub-dimension. This yields groups $\{l_g, \hat{a}_g^{(1)}, \hat{a}_g^{(2)}\}$, where the two variants may be associated with different sampled sub-dimensions.

**Verification and filtering.** We verify each variant using DeepSeek-V3 (Liu et al., 2024) along two axes: (i) anomaly invariance (no new/missing anchored events; no key outcome changes) and (ii) constraint satisfaction (the sampled sub-dimension is realized). Failed generations are retried up to 3 times; groups that still fail verification are discarded. We further conduct manual checking to confirm both anomaly invariance and correct constraint realization. After filtering, CVP contains $G = 898$ valid groups, and the distribution of selected dimensions is shown in Fig. 5. Complete prompt details, examples, and statistics can be found in Appendix C.

## 4. Class-Guided Chain-of-Evaluation

### 4.1. Problem setup, symbols, and output

Given a $(\hat{a}, l)$, CG-CoE outputs event-level Precision/Recall and a match relation:

$$\text{CG-CoE}(\hat{a}, l) \rightarrow \left(\text{Prec, Rec, } \mathcal{E}^{\hat{a}}, \mathcal{E}^l, M\right). \quad (7)$$

Here Prec measures the fraction of prediction-side anomalous events that are supported, and Rec measures how completely the prediction covers label-side anomalous events. $\mathcal{E}^{\hat{a}}$ and $\mathcal{E}^l$ are the anomalous-event sets extracted in Step 1 (each event stored with a verbatim evidence span and an evidence-anchored rewrite). $M \subseteq \mathcal{E}^{\hat{a}} \times \mathcal{E}^l$ is the bipartite support relation constructed in Step 2.

### 4.2. Method

#### 4.2.1. STEP 1: MINIMAL EVENT EXTRACTION AND REWRITES

CG-CoE extracts a de-duplicated set of minimal anomalous events from each text. We use minimal granularity: each event corresponds to one core abnormal act or outcome; if multiple abnormal acts/outcomes appear in one sentence, we split them into separate events. This step outputs $\mathcal{E}^{\hat{a}}$ and $\mathcal{E}^l$.

Each event is recorded with (i) a verbatim evidence span and (ii) an evidence-anchored rewrite derived from that span. The rewrite must be strictly supported by its evidence span and cannot introduce new information.

**Label-side per-event class hints.** For each label event in $\mathcal{E}^l$, CG-CoE also produces a lightweight `class_hint` (JSON field). It is used only in Stage B (Step 2) to gate candidates and bound the within-class tolerance boundary, and is not treated as matching evidence.

#### 4.2.2. STEP 2: STRICT-THEN-FLEXIBLE MATCHING WITH CLASS-GUIDED TOLERANCE BOUNDARY

CG-CoE builds a bipartite support relation $M \subseteq \mathcal{E}^{\hat{a}} \times \mathcal{E}^l$, where an edge $(e, e') \in M$ indicates that prediction event $e$ provides evidence-backed support for label event $e'$. To accommodate split/merge effects from Step 1, a label event may be supported by multiple prediction events (compositional support).

Matching proceeds in two stages. Stage A adds high-confidence edges by strict evidence alignment. Stage B is applied only to label events that remain unsupported after Stage A; `class_hint` constrains the candidate space and admissible tolerance, while adding an edge still requires evidence-backed support.

**Stage A (Strict): Direct evidence-based matching.** Stage A matches events using only the verbatim evidence spans (not rewritten statements). We add an edge $(e, e')$ iff the two spans entail the same minimal abnormal act/outcome with consistent key arguments (e.g., who did what to whom/what, and the outcome), allowing paraphrases and minor reordering.

**Stage B (Flexible, bounded): Class-guided matching.** For each unmatched label event $e' \in \mathcal{E}^l$, we form a gated candidate set from $\mathcal{E}^{\hat{a}}$ using `class_hint`. A prediction event $e$ is admitted if its evidence provides concrete support for the same core abnormal act/outcome, allowing partial matching when the correspondence arises from event split or merge, contains no explicit conflict in the core act/outcome or essential role assignments, and stays within the anomaly

*Table 3.* **Meta-evaluation on AEA and CVP.** AEA reports Spearman's rank correlation with human event-level Precision/Recall. CVP reports mean within-group score variation $\mathbb{E}[\Delta]$ per perturbation dimension (lower is better). **Avg.** is the unweighted mean over the five CVP dimensions.

| | AEA $\rho \uparrow$ | | CVP $\mathbb{E}[\Delta] \downarrow$ (by dimension) | | | | | |
|---|---|---|---|---|---|---|---|---|
| Metrics | $\rho_{\mathrm{Prec}}$ | $\rho_{\mathrm{Rec}}$ | Linguistic Style _Style | Narrative _Strategy | Information _Structure | Lexical _Reference | Discourse _Structure | **Avg.** |
| **N-gram-based** | | | | | | | | |
| BLEU | 0.230 | 0.249 | 0.049 | 0.047 | 0.032 | 0.031 | 0.025 | 0.037 |
| ROUGE-L | 0.362 | 0.378 | 0.062 | 0.058 | 0.053 | 0.047 | 0.046 | 0.053 |
| **Embedding-based** | | | | | | | | |
| BERTScore | 0.407 | 0.425 | 0.041 | 0.039 | 0.037 | 0.032 | 0.030 | 0.036 |
| BARTScore | 0.313 | 0.319 | 0.009 | 0.009 | 0.008 | 0.008 | 0.004 | 0.008 |
| UniEval | 0.633 | 0.634 | 0.108 | 0.107 | 0.097 | 0.097 | 0.090 | 0.100 |
| **LLM-based** | | | | | | | | |
| HAWK *(consistency)* | 0.710 | 0.710 | 0.115 | 0.100 | 0.095 | 0.092 | 0.092 | 0.099 |
| HAWK *(reasonability)* | 0.706 | 0.709 | 0.117 | 0.100 | 0.093 | 0.089 | 0.088 | 0.097 |
| VAD-R1 *(classification)* | 0.731 | 0.735 | 0.091 | 0.087 | 0.084 | 0.079 | 0.071 | 0.083 |
| VAD-R1 *(action_matching)* | 0.700 | 0.704 | 0.104 | 0.090 | 0.088 | 0.086 | 0.068 | 0.087 |
| VAD-R1 *(factual_consistency)* | 0.722 | 0.726 | 0.096 | 0.090 | 0.088 | 0.084 | 0.068 | 0.085 |
| CG-CoE (Precision) | **0.800** | 0.785 | 0.087 | **0.066** | 0.051 | 0.051 | 0.038 | 0.059 |
| CG-CoE (Recall) | 0.790 | **0.813** | **0.085** | 0.068 | **0.046** | **0.030** | **0.028** | **0.051** |

class specified by `class_hint`, thereby disallowing cross-class substitutions.

If no single candidate sufficiently supports $e'$, we allow compositional support by selecting a minimal supporting set of gated prediction events and adding an edge from each selected event to $e'$ in $M$.

#### 4.2.3. STEP 3: SCORING WITH EVENT-LEVEL PRECISION AND RECALL

Given $\mathcal{E}^{\hat{a}}$, $\mathcal{E}^l$, and $M$, we compute event-level Precision/Recall using the same counting convention and edge-case handling as AEA (Sec. 3.2.1). A prediction event contributes at most once to Precision regardless of how many label events it supports.

### 4.3. Determinism and verifiability

We instantiate the CG-CoE framework using DeepSeek-V3 (Liu et al., 2024) as the backbone LLM. To ensure rigorous evaluation and system stability, we employ a strictly defined JSON schema and a fixed prompt template. The comprehensive prompt and schema specifications are provided in Appendix D.

## 5. Experiments

**Meta-evaluation benchmarks.** We meta-evaluate metrics on two benchmarks: AEA for validity and CVP for robustness (see Sec. 3.2 and Sec. 3.3 for more details).

**Baselines.** We compare CG-CoE with three families of representative metrics to assess whether existing baselines are suitable for evaluating VAU models' anomaly understanding ability. N-gram-based metrics include BLEU (Pap-

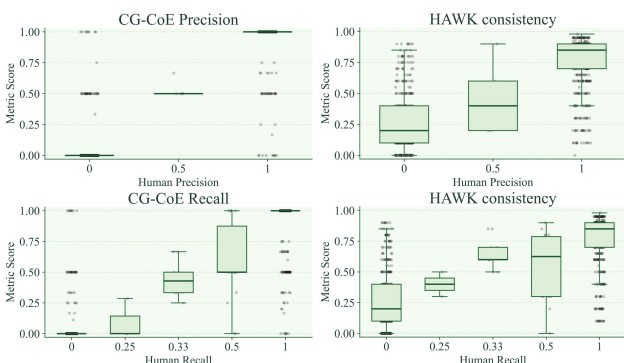

*Figure 6.* **Score distributions on AEA conditioned on human event-level Precision and Recall.** Score distributions on AEA conditioned on human event-level Precision (top) and Recall (bottom). The x-axis bins samples by the corresponding human score, and the y-axis shows metric scores. Boxplots indicate median and interquartile range; points are individual samples.

ineni et al., 2002) and ROUGE-L (Lin, 2004). Embedding-based metrics include BERTScore (Zhang et al., 2019) , BARTScore (Yuan et al., 2021) and UniEval (Zhong et al., 2022). We compute BERTScore using `roberta-large` and instantiate BARTScore with `bart-large-cnn`. LLM-based metrics include VAU-R1 (Zhu et al., 2025a) and HAWK (Tang et al., 2024).

For multi-dimensional VAU metrics, we do not report all components. Instead, we select dimensions most directly related to anomaly understanding. For HAWK, we report `consistency` and `reasonability`. For VAU-R1, we report `classification`, `action_matching`, and `factual_consistency`. Some metrics do not naturally fall within the [0, 1] range, which is crucial for fair comparison on CVP. For metrics with a well-defined bounded range, we continue to apply a linear rescaling to [0, 1]; for

unbounded metrics such as BARTScore, we instead use a log-based mapping for normalization.

## 5.1. Main Results

Table 3 summarizes the meta-evaluation results of all metrics on AEA and CVP, showing a clear trade-off between validity and robustness.

**Validity on AEA.** CG-CoE agrees best with human event-level judgments, achieving the highest Spearman correlations on both targets: $\rho_{\text{Prec}}=0.800$ and $\rho_{\text{Rec}}=0.813$. Compared with the strongest LLM-based baseline, CG-CoE improves validity by $+0.069$ on $\rho_{\text{Prec}}$ and $+0.078$ on $\rho_{\text{Rec}}$ (about 9–11% relative). In contrast, embedding-based metrics correlate much weaker with event-level labels, suggesting that lexical overlap or generic semantic similarity cannot reliably capture whether anomalous events are hallucinated or missed. Figure 6 supports this: as $\text{Prec}^h$ or $\text{Rec}^h$ increases, CG-CoE scores move upward with less overlap across bins, while HAWK shows larger spread and more overlap, indicating noisier alignment with partial event correctness.

**Robustness on CVP.** CVP measures robustness by the mean within-group variation $\mathbb{E}[\Delta]$ under controlled stylistic perturbations. Among LLM-based metrics, CG-CoE has the lowest variation, reducing $\mathbb{E}[\Delta]$ by 29%–39% compared to the best LLM baseline, and also outperforming HAWK. This result is consistent across perturbation dimensions; *Linguistic Style* is the most sensitive dimension for all methods, indicating that style-only rewrites remain the hardest setting. Figure 7 gives a distribution-level view: CG-CoE assigns most pairs to $|\Delta|=0$ (e.g., 840/837 exact zeros for the Precision/Recall views) and produces few large changes (e.g., $|\Delta|>0.3$: 35/28), whereas HAWK assigns much more mass to non-zero bins. Notably, BARTScore has very low variation but poor AEA validity, showing that robustness alone can come from an overly insensitive metric and does not ensure anomaly-event faithfulness.

**Overall.** Overall, CG-CoE achieves the best event-level validity on AEA and remains stable under anomaly-preserving stylistic variations on CVP, enabling more reliable comparison of VAU models' anomaly understanding rather than their descriptive style.

## 5.2. Ablation on CG-CoE.

Table **??** presents an ablation study of the core CG-CoE design on our meta-benchmark, covering AEA (validity) and CVP (robustness). For AEA, we report $\rho_{\text{Prec}}$ and $\rho_{\text{Rec}}$.

**(i) Anomalous event extraction** yields the strongest robustness gain. By filtering normal content, extracting minimal

| Setting | AEA ↑ | | CVP ↓ |
|---|---|---|---|
| | $\rho_{\text{Prec}}$ | $\rho_{\text{Rec}}$ | Avg. |
| Baseline (P) | 0.682 | 0.682 | 0.118 |
| Baseline (R) | 0.677 | 0.679 | 0.122 |
| + Extract anomalous events (P) | 0.727 | 0.724 | 0.072 |
| + Extract anomalous events (R) | 0.730 | 0.735 | 0.074 |
| + Compositional Match (P) | 0.772 | 0.764 | 0.063 |
| + Compositional Match (R) | 0.769 | 0.782 | 0.061 |
| CG-CoE (P) | **0.800** | 0.785 | 0.059 |
| CG-CoE (R) | 0.790 | **0.813** | **0.051** |

*Table 4.* **Ablation results on AEA and CVP.**

| LLM | AEA $\rho$ ↑ | CVP $\mathbb{E}[\Delta]$ ↓ |
|---|---|---|
| Qwen3.5-0.8B | N/A | N/A |
| Qwen3.5-2B | N/A | N/A |
| Qwen3.5-4B | N/A | N/A |
| Qwen3.5-9B | 0.861 | 0.121 |
| Qwen3.5-27B | 0.858 | 0.110 |
| Qwen3.5-35B | 0.871 | 0.083 |
| Qwen3.5-122B | 0.853 | 0.092 |
| Qwen3.5-197B | 0.862 | 0.070 |

*Table 5.* **Performance metrics for different Qwen3.5 model sizes.**

events, and rewriting into a uniform style, it reduces sensitivity to various expression styles and normal content, leading to a drop in CVP. It also improves validity by tying scores more directly to anomaly evidence.

**(ii) Compositional matching** further improves validity and robustness by mitigating granularity mismatch between predictions and labels (e.g., merged or fragmented events). Flexible aggregation yields more faithful event-level alignment and stabilizes scores under CVP perturbations that repackage the same content with different structure. This is especially important for long or multi-event answers, where structural rewrites can change segmentation without changing anomaly semantics.

**(iii) Class-guided tolerance boundary** provides additional gains by constraining flexible matching to class-relevant semantics, reducing alignments between different classes. This variant achieves the best AEA correlations and the lowest CVP variation overall. The trend suggests that bounding semantic flexibility by class is crucial when recovering missing label events while avoiding mismatched substitutions.

## 5.3. Ablation on LLM Scale

Table 5 summarizes the performance of different model scales within the Qwen3.5 family on a set of 100 samples. Models ranging from 0.8B to 4B are marked as N/A, as

these smaller models often generated severely corrupted or uninterpretable outputs, making their evaluations unreliable. This observation highlights that the effectiveness of CG-CoE depends on the judge LLM's ability to follow instructions and perform structured semantic reasoning.

The results further reveal a threshold effect rather than a simple monotonic scaling trend. Once the model reaches a sufficient capability level—around 9B and above—the intermediate reasoning steps stabilize, allowing CG-CoE to operate reliably across different backbones within this regime. In other words, CG-CoE is not specific to any single model, such as DeepSeek; instead, it requires an LLM capable of consistent instruction adherence and structured reasoning.

### 5.4. Ablation on LLM Backbones

We further compare metrics by using the same backbone across all baseline models. Tables 6 and 7 report performance on the AEA and CVP metrics, respectively. While absolute scores vary depending on the backbone, CG-CoE consistently outperforms others, confirming that the advantage stems from the method itself rather than a particular model choice.

| Model↑ | CG-CoE | HAWK | VAD-R1 |
|---|---|---|---|
| Claude-Sonnet-4-6 | **0.751** | 0.700 | 0.720 |
| DeepSeek-V3.2 | **0.766** | 0.688 | 0.655 |
| Gemini-3.1 | **0.751** | 0.682 | 0.719 |
| GPT-5.4 | **0.718** | 0.685 | 0.701 |

*Table 6.* **Comparison across different backbone models on AEA.**

| Model↓ | CG-CoE | HAWK | VAD-R1 |
|---|---|---|---|
| Claude-Sonnet-4-6 | **0.023** | 0.044 | 0.034 |
| DeepSeek-V3.2 | **0.027** | 0.045 | 0.050 |
| Gemini-3.1 | **0.031** | 0.052 | 0.034 |
| GPT-5.4 | **0.035** | 0.040 | 0.042 |

*Table 7.* **Comparison across different backbone models on CVP.**

### 6. Limitations

Our current metric assumes that the reference contains sufficient anomalous-event information for semantic evaluation. Under this setting, the evaluator can assess whether anomalous events are correctly understood by comparing the prediction with the reference at the event level. However, when the reference is too incomplete or underspecified to support reliable judgment, this text-only setting becomes inherently limited. In such cases, accurate evaluation may require direct access to the video content rather than relying solely on the textual reference.

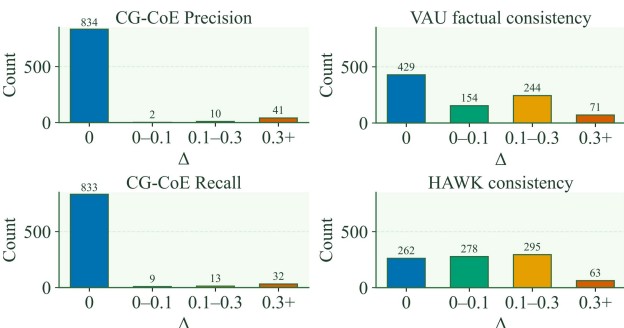

*Figure 7.* **Binned distributions of score variation on the CVP benchmark.** $|\Delta|$ is grouped into four ranges: 0, 0–0.1, 0.1–0.3, and 0.3+, where more mass near 0 indicates stronger robustness to stylistic perturbations.

In addition, the current benchmark is mainly constructed based on Holmes-VAU. Although this setting is sufficient to support our initial meta-evaluation study, broader validation across more VAU datasets with different anomaly categories, narrative styles, and annotation protocols would further strengthen the generality of our conclusions.

### 7. Conclusion

In summary, we introduce CG-CoE and an anomaly-focused meta-evaluation benchmark to jointly assess validity and robustness for VAU metrics under style and normal-content confounders. Our results show that evidence-grounded event extraction and class-guided matching substantially improve reliable metric validation, enabling fairer VAU model comparison.

### Acknowledgements

This work was supported in part by the National Natural Science Foundation of China under Grants No. 62221005, 62472060, U22A2096 and U23A20318, in part by the Science and Technology Innovation Key R&D Program of Chongqing under Grant No. CSTB2023TIAD-STX0016, in part by the Natural Science Foundation of Chongqing under Grants No. CSTB2024NSCQ-QCXMX0060.

### Impact Statement

This work aims to advance trustworthy evaluation for Video Anomaly Understanding by improving metric validity, robustness, and verifiability. As VAU systems can be used in monitoring settings, responsible use should follow applicable privacy and consent requirements, and automatic evaluation should serve as a verifiable aid rather than the sole basis for deployment decisions.

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

# A. Summary of Appendix

This appendix complements the main paper with (i) the full human annotation protocol for AEA, (ii) the construction details of CVP for robustness diagnosis, and (iii) the exact prompt used by CG-CoE.

**AEA: annotation manual and reproducibility (§B).**    We provide the complete guideline for deriving human event-level Precision/Recall $(\mathrm{Prec}^h, \mathrm{Rec}^h)$ from each prediction–label pair $(\hat{a}, l)$. Annotators (1) extract de-duplicated minimal anomalous events from both texts with contiguous evidence spans and anomaly classes, (2) build a bipartite match relation that requires incident-meaning agreement and aligned evidence (allowing many-to-many compositional support under split/merge), and (3) compute $(\mathrm{Prec}^h, \mathrm{Rec}^h)$ with conservative handling of empty-set cases (§B.2–§B.5). We also describe how events, evidence, classes, and the final matching $M^h$ are stored to support adjudication and traceability (§B.6).

**Generation details and data curation (§B.7).**    We report the VAU models used to produce predictions (Holmes-VAU, HAWK, Qwen3-VL-Think-30B, Qwen3-VL-Instruct-30B, Qwen2.5-VL-7B, InternVL2.5-8B), unified decoding settings (deterministic decoding with $T{=}0$, fixed seed, and a 4096-token budget), and hardware ($8\times$ RTX A6000 48GB) to ensure reproducibility and comparability across models. We further document the data cleaning outcome: 32 instances are discarded due to responses irrelevant to the corresponding videos.

**AEA prompts, interface, and example (Fig. 8–Fig. 10).**    We include the question prompt used to collect/standardize answers (Fig. 8), the annotation platform for event extraction and matching (Fig. 9), and a fully annotated example (Fig. 10).

**CVP: prompts, edit taxonomy, failures, and detailed experiments (§C and §C.3).**    We provide the prompts used to build CVP, including anomaly anchor extraction (Fig. 11), variant checking (Fig. 13), and variant generation (Fig. 14), together with a worked example of controlled variant generation (Fig. 12). We report the predefined edit taxonomy and its rules under a global constraint that anomalous semantics are fixed by ANOMALY_ANCHOR and edits are restricted to NORMAL content (Table 8), and note observed failures (2 cases in Referential_Explicitness). Finally, we present dimension-wise robustness results on CVP via within-group score variation $\mathbb{E}[\Delta]$ for each controlled sub-dimension (Table 9).

**CG-CoE prompt (Fig. 15).**    For completeness and reproducibility, we include the exact prompt used by CG-CoE in Fig. 15.

# B. AEA

## B.1. Annotation Manual for AEA: Anomalous Event-level Precision/Recall

This appendix details the human annotation protocol used to derive event-level Precision/Recall $(\mathrm{Prec}^h, \mathrm{Rec}^h)$ for AEA (§3.2). Given a prediction–label pair $(\hat{a}, l)$, annotators (i) extract evidence-backed anomalous events from both texts, (ii) match events by incident meaning with aligned evidence, and (iii) compute $(\mathrm{Prec}^h, \mathrm{Rec}^h)$ from the resulting match relation. The protocol is designed to (a) penalize hallucinated anomalies (precision) and (b) penalize missing anomalies (recall), while preventing *type-only* credit without evidence alignment.

## B.2. Notation and annotation outputs

For each $(\hat{a}, l)$, annotators produce: (i) anomalous-event sets $\mathcal{E}^{\hat{a}}$ and $\mathcal{E}^l$; (ii) a bipartite match relation $M^h \subseteq \mathcal{E}^{\hat{a}} \times \mathcal{E}^l$; and (iii) the derived $(\mathrm{Prec}^h, \mathrm{Rec}^h)$. Each extracted event is associated with an anomaly class and a supporting evidence span from the source text.

## B.3. Step 1: Extract anomalous events with evidence

Annotators extract anomalous events independently from $l$ and $\hat{a}$, forming $\mathcal{E}^l$ and $\mathcal{E}^{\hat{a}}$. An extracted event is the minimal unit describing a concrete abnormal incident (action/outcome/state change). For each event $e$, annotators record: (i) an anomaly class $\mathrm{class}(e)$ from the predefined label set; and (ii) an evidence span $\mathrm{ev}(e)$—a short contiguous span in the source text that supports the event.

**Evidence-backed inclusion.** An event is recorded *only if* the incident evidence supports that the abnormal incident occurred. Statements that are purely speculative (e.g., "might be"), normative, or generic without occurrence evidence are not recorded.

**No class-only events.** Mentions that only state an anomaly type/class without incident evidence ("it is theft") are ignored. Concretely, class agreement alone is insufficient: an event must be grounded by an evidence span describing the abnormal act/outcome.

**De-duplication and minimality.** Paraphrases referring to the same incident are de-duplicated on each side. Annotators aim for minimal, incident-distinguishing events: multiple independent abnormal incidents should be split into separate events when the text provides separable evidence; conversely, tightly coupled descriptions may remain a single event when they cannot be reliably separated by evidence.

**Negation and self-contradiction.** Explicitly negated anomalies are not recorded. If the text contradicts itself regarding occurrence, annotators default to not recording the event unless a clear, final occurrence claim is evidence-backed.

### B.4. Step 2: Match events by incident meaning and aligned evidence

Annotators construct a bipartite relation $M^h \subseteq \mathcal{E}^{\hat{a}} \times \mathcal{E}^l$. A pair $(e, e') \in M^h$ is added *iff* $e \in \mathcal{E}^{\hat{a}}$ and $e' \in \mathcal{E}^l$ describe the *same abnormal incident*. We require the following:

**Incident meaning agreement.** The core abnormal act/outcome described by $e$ and $e'$ must be consistent (allowing paraphrase and granularity differences). Missing optional details are treated as lack of evidence rather than contradiction. A contradiction requires an explicit disagreement about the core act/outcome.

**Class consistency.** $\text{class}(e)$ and $\text{class}(e')$ must be consistent. If the class taxonomy defines equivalence or allowable within-group mappings, annotators follow that mapping; otherwise, strict class agreement is required.

**Evidence alignment (required).** In addition to class consistency, annotators must identify evidence spans $\text{ev}(e)$ and $\text{ev}(e')$ that refer to the same incident. **Class agreement without aligned evidence does not constitute a match.**

**Compositional support.** To accommodate event split/merge during extraction, we allow many-to-many matching: a label-side event may match multiple prediction-side events when they jointly support the same incident (compositional support). Similarly, a prediction-side event may match multiple label-side events only when evidence indicates they refer to the same incident at different granularities.

### B.5. Step 3: Compute human Precision/Recall

Let $n_{\hat{a}} = |\mathcal{E}^{\hat{a}}|$ and $n_l = |\mathcal{E}^l|$. Define $n_{\hat{a}}^m$ as the number of prediction-side events that participate in at least one pair in $M^h$, and $n_l^m$ as the number of label-side events that participate in at least one pair in $M^h$:

$$n_{\hat{a}}^m = \left|\left\{e \in \mathcal{E}^{\hat{a}} \mid \exists e' \in \mathcal{E}^l, \, (e, e') \in M^h\right\}\right|, \quad n_l^m = \left|\left\{e' \in \mathcal{E}^l \mid \exists e \in \mathcal{E}^{\hat{a}}, \, (e, e') \in M^h\right\}\right|.$$

When $n_{\hat{a}} > 0$ and $n_l > 0$, we compute:

$$\text{Prec}^h = \frac{n_{\hat{a}}^m}{n_{\hat{a}}}, \qquad \text{Rec}^h = \frac{n_l^m}{n_l}.$$

We handle empty-set edge cases conservatively: if $\mathcal{E}^{\hat{a}} = \mathcal{E}^l = \emptyset$, we set $(\text{Prec}^h, \text{Rec}^h) = (1, 1)$; otherwise, if exactly one side is empty, we set $(\text{Prec}^h, \text{Rec}^h) = (0, 0)$.

### B.6. Implementation notes for reproducibility

To support reproducibility and adjudication, annotators store extracted events with their evidence spans and anomaly classes, as well as the final match relation $M^h$. This enables downstream inspection of disagreement sources (event extraction vs. matching) and ensures the human scores are traceable to textual evidence.

You are given a video. Your task is:
1. Identify whether the video contains an anomaly.
2. If there is an anomaly, classify it into a specific anomaly type (e.g., arrest, abuse, assault, violence, burglary, explosion, fighting, shooting, shoplifting, stealing, vandalism etc.).
3. Provide a concise description of the anomaly, focusing only on the abnormal action/event that makes it anomalous.
4. If there is no anomaly, clearly state that no anomaly is present and briefly describe the main normal activity in the video.

*Figure 8.* **Question prompt used for AEA annotation.**

### B.7. Generation Details

**VAU models.**  We include predictions from the following VAU models: HOLMES-VAU, HAWK, QWEN3-VL-THINK-30B, QWEN3-VL-INSTRUCT-30B, QWEN2.5-VL-7B, and INTERNVL2.5-8B. For all models, we use a unified prompting format (same question and context) and keep other generation settings consistent to ensure comparability.

**Decoding and token budget.**  Unless otherwise specified, we use deterministic decoding (greedy) with temperature $T = 0$ and a fixed random seed for reproducibility. We set the maximum context length to $4096$ tokens; inputs exceeding this budget are truncated to fit the limit.

**Hardware.**  All experiments are run on a single machine with $8\times$ NVIDIA RTX A6000 GPUs (48GB VRAM each).

### B.8. Discarded Instances

After verification, we discarded 32 instances in total (from HAWK and Holmes-VAU), all because the responses were irrelevant to the corresponding videos.

### B.9. AEA Prompts, Interface, and Example

We provide the question prompt in Fig. 8, the annotation platform in Fig. 9, and an annotated example in Fig. 10.

## C. CVP

### C.1. Prompts and Example

We show the anchor extraction prompt in Fig. 11, the variant check prompt in Fig. 13, and the variant generation prompt in Fig. 14. An example of CVP variant generation is provided in Fig. 12.

### C.2. Edit Dimensions and Failures

The predefined taxonomy and sampling statistics are reported in Table 8.  We observed 2 failures in the `Referential_Explicitness` dimension.

### C.3. Detailed Experiments

The dimension-wise robustness results on CVP (i.e., within-group score variation $\mathbb{E}[\Delta]$ for each controlled sub-dimension) are reported in Table 9.

## D. CG-CoE

Prompt of CG-CoE is shown in Fig. 15.

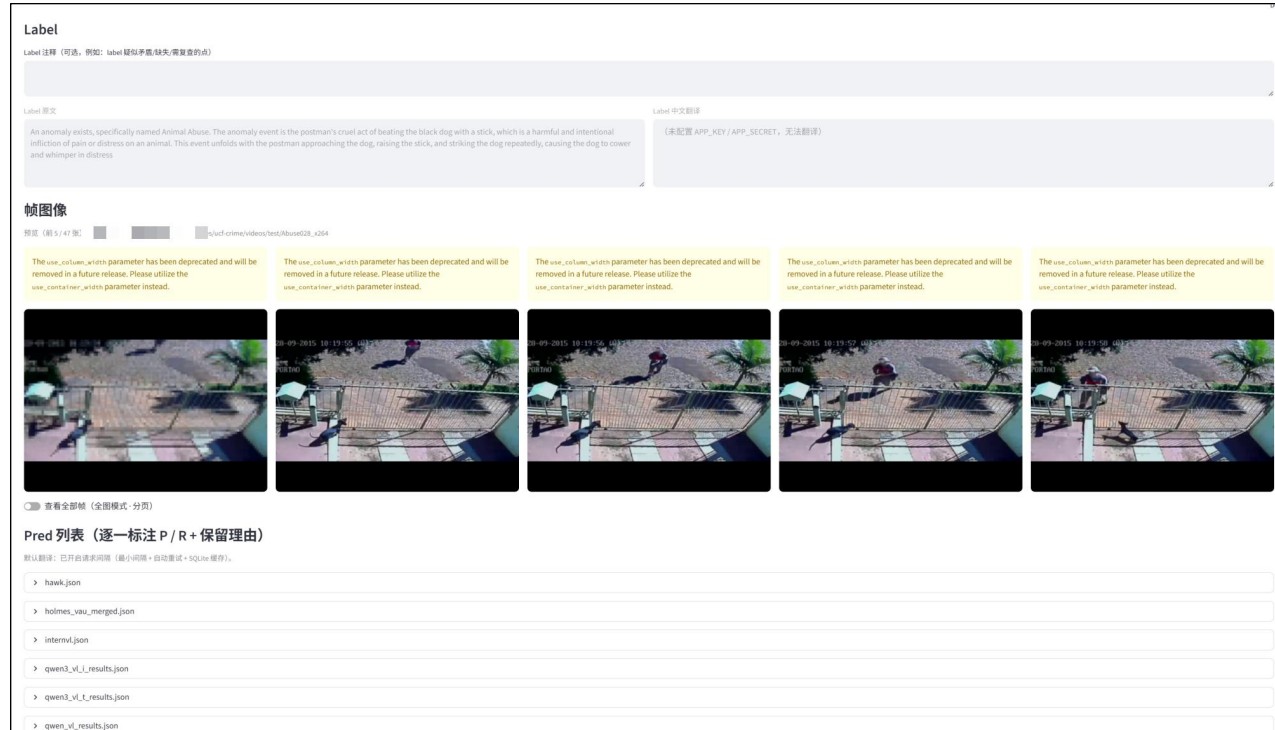

*Figure 9.* **Annotation platform for AEA.**

*Table 8.* **Controlled-variant dimension rules. Global constraint:** anomalous events and semantics are fixed by `ANOMALY_ANCHOR`; all edits are restricted to **NORMAL** content unless explicitly stated.

| Dimension | Num | Level 0 | Level 2 |
|---|---|---|---|
| **A) Linguistic Style** | | | |
| Formality | 48 | Informal tone; contractions; shorter sentences. | Formal tone; no contractions; more formal connectives. |
| Objectivity | 102 | Mild subjective wording allowed for NORMAL only. | Objective/impersonal reporting; observable-only. |
| Point_of_View | 102 | Third-person / camera framing. | First-person *observer* only; not a participant. |
| Sentiment | 66 | Mild negative tint for NORMAL ambiance. | Mild positive tint for NORMAL ambiance. |
| Modal_Strength | 90 | Hedge NORMAL statements (might/could/appears). | More assertive for NORMAL without fabricating certainty. |
| **B) Information Structure** | | | |
| Granularity | 60 | Coarse NORMAL summary (1–2 broad statements). | Fine-grained NORMAL steps/actions (more decomposed). |
| Detail_Density | 60 | Low NORMAL background detail; remove extras. | Add safe generic NORMAL details; no names/numbers. |
| Length | 96 | Overall text shortened; anomaly content unchanged. | Overall text expanded via NORMAL-only background. |
| Redundancy | 78 | No repetition; each NORMAL point stated once. | Exactly one paraphrase of a NORMAL point; no anomaly repetition. |
| Summarization | 30 | Abstract NORMAL phrasing (high-level). | Expanded NORMAL with generic categories (no names/numbers). |
| **C) Discourse Structure** | | | |
| Temporal_Order | 30 | NORMAL described chronologically. | Reverse/flashback framing for NORMAL only; anomaly order unchanged. |
| Event_Order_Clarity | 78 | Loose sequencing; minimal connectors. | Explicit connectors (first/then/afterward/meanwhile). |
| Paragraphing | 108 | Single paragraph (no bullets/blank lines). | Bullet/numbered list; structure-only change. |

| Dimension | # | Level 0 | Level 2 |
|---|---|---|---|
| Cohesion | 60 | Minimal discourse markers. | Add explicit discourse markers (additionally/consequently/etc.). |
| **D) Lexical Reference** | | | |
| Lexical_Paraphrase | 72 | Minimal paraphrase; wording close to original. | Strong paraphrase + restructuring for NORMAL only. |
| Referential_Explicitness | 48 | Prefer pronouns where unambiguous. | Prefer explicit noun phrases (the man/driver/pedestrian). |
| Quantitative_Precision | 54 | Approximate quantifiers for NORMAL; keep existing numbers. | Preserve existing numbers/times only; introduce no new numbers. |
| SpatialTemporal_Specificity | 102 | Generic NORMAL spatial/temporal phrases. | More explicit only if already present; no new time/location. |
| Concreteness | 84 | Abstract wording for NORMAL (high-level verbs/nouns). | Concrete visual wording for NORMAL (more depictive). |
| **E) Narrative Strategy** | | | |
| Narrativity | 72 | Report-like, minimal storytelling. | Mild story-like flow phrases; no invented dialogue/thoughts. |
| Context_Background | 66 | Minimal NORMAL background. | Add safe generic NORMAL background (no incidents/names). |
| Focus | 138 | Anomaly-centric; NORMAL kept brief. | Before/after framing using only implied/available info. |
| Normal_Ratio | 90 | Low NORMAL share (very brief background). | High NORMAL share (NORMAL dominates; still normal). |
| Anomaly_Position | 66 | Place anomaly mentions early in the text. | Place anomaly mentions late after NORMAL buildup. |

```
{  "id": "60",
   "label": "The anomaly exists and is specifically identified as a
Riot anomaly. The anomaly event involves chaotic scenes of
protests and physical altercations between police officers and
protesters, characterized by the presence of riot gear, smoke,
broken glass, and signs of destruction, which unfold from start
to end with intense and violent clashes, property damage, and a
general atmosphere of unrest and turmoil",
   "label_note": "Riots primarily involve scenes of confrontation
and chaos.",
   "pred": "The video depicts a nighttime urban scene with people
walking and interacting on a street. There are police officers
present, some of whom appear to be wearing riot gear. The
environment suggests a protest or demonstration.Anomaly
Classification: Type: Violence
Description:The presence of tear gas canisters being deployed by
law enforcement indicates an act of violence. Tear gas is
typically used as a means of crowd control and dispersal, which
is considered an aggressive and violent action in this context.
The smoke from the tear gas canisters is visible, and people are
seen reacting to it, suggesting that they are being affected by
the gas. This is an unusual and concerning event for a peaceful
protest setting.",
   "P": 1.0,   "R": 1.0,   "keep_reason": "While it is a
misclassification, it effectively captures the essence of the
anomaly.",   "source_file": "qwen_vl_results.json"
 }
```

*Figure 10.* **Example of AEA annotation.**

```
You are a strict extractor. Given a VAU model answer text (pred),
extract a canonical, stable anomaly anchor.
Output JSON ONLY. No markdown, no code fences, no extra
text.
Rules:- Be conservative: do NOT hallucinate anomalies.
   - If pred indicates normal/no-incident, set has_anomaly=false
and keep lists empty.
   - If has_anomaly=true, keep anomaly summary minimal but
unambiguous.
   - Keep wording stable and generic (avoid stylistic variation).
   JSON schema:
   {
     "has_anomaly": true/false,
     "anomaly_types": [string, ...],
     "anomaly_summary": string,
     "anomaly_events": [
       {
         "event": string,
         "who": string,
         "action": string,
         "object_or_target": string,
         "where_when": string,
         "outcome": string
       }, ...
     ]
   }
```

*Figure 11.* **Prompt for anomaly anchor extraction.**

*Figure 12.* **Example of CVP variant generation.**

*Figure 13.* **Prompt for CVP variant checking.**

*Table 9.* **Dimension-wise within-group score variation $\mathbb{E}[\Delta]$ on CVP (lower is better).** Column abbreviations: Len=Length, LexP=Lexical Paraphrase, Summ=Summarization, Conc=Concreteness, Obj=Objectivity, AnoPos=Anomaly Position, NormR=Normal Ratio, DetDen=Detail Density, Sent=Sentiment, Quant=Quantitative Precision, TempO=Temporal Order, STSpec=SpatialTemporal Specificity, POV=Point of View, CtxBg=Context/Background, Coh=Cohesion, Foc=Focus, Red=Redundancy, Form=Formality, Gran=Granularity, EOC=Event Order Clarity, RefExp=Referential Explicitness, Narr=Narrativity, Para=Paragraphing, Mod=Modal Strength.

| Metric | Len | LexP | Summ | Conc | Obj | AnoPos | NormR | DetDen | Sent | Quant | TempO | STSpec | POV | CtxBg | Coh | Foc | Red | Form | Gran | EOC | RefExp | Narr | Para | Mod |
|---|---|---|---|---|---|---|---|---|---|---|---|---|---|---|---|---|---|---|---|---|---|---|---|---|
| BLEU | 0.077 | 0.076 | 0.070 | 0.067 | 0.052 | 0.051 | 0.046 | 0.044 | 0.043 | 0.040 | 0.038 | 0.036 | 0.033 | 0.032 | 0.032 | 0.031 | 0.025 | 0.024 | 0.024 | 0.024 | 0.020 | 0.016 | 0.016 | 0.014 |
| ROUGE | 0.098 | 0.091 | 0.068 | 0.075 | 0.068 | 0.081 | 0.094 | 0.045 | 0.056 | 0.055 | 0.068 | 0.048 | 0.051 | 0.045 | 0.045 | 0.047 | 0.037 | 0.045 | 0.035 | 0.040 | 0.037 | 0.038 | 0.040 | 0.026 |
| BERTScore | 0.174 | 0.146 | 0.134 | 0.147 | 0.119 | 0.147 | 0.156 | 0.113 | 0.098 | 0.093 | 0.103 | 0.074 | 0.100 | 0.085 | 0.101 | 0.079 | 0.079 | 0.116 | 0.062 | 0.071 | 0.041 | 0.070 | 0.064 | 0.059 |
| BARTScore | 0.018 | 0.012 | 0.012 | 0.015 | 0.014 | 0.025 | 0.018 | 0.008 | 0.021 | 0.004 | 0.008 | 0.006 | 0.007 | 0.006 | 0.004 | 0.005 | 0.005 | 0.007 | 0.004 | 0.005 | 0.003 | 0.003 | 0.003 | 0.003 |
| UniEval | 0.134 | 0.106 | 0.106 | 0.133 | 0.130 | 0.108 | 0.161 | 0.093 | 0.057 | 0.099 | 0.109 | 0.065 | 0.139 | 0.085 | 0.073 | 0.077 | 0.107 | 0.134 | 0.083 | 0.075 | 0.093 | 0.097 | 0.105 | 0.077 |
| HAWK-Reason | 0.096 | 0.058 | 0.078 | 0.105 | 0.100 | 0.123 | 0.162 | 0.102 | 0.107 | 0.089 | 0.110 | 0.086 | 0.083 | 0.110 | 0.108 | 0.106 | 0.111 | 0.088 | 0.103 | 0.082 | 0.119 | 0.112 | 0.087 | 0.076 |
| HAWK-Detail | 0.160 | 0.092 | 0.164 | 0.145 | 0.117 | 0.162 | 0.218 | 0.149 | 0.116 | 0.089 | 0.119 | 0.094 | 0.114 | 0.132 | 0.128 | 0.105 | 0.096 | 0.115 | 0.097 | 0.101 | 0.116 | 0.109 | 0.121 | 0.094 |
| HAWK-Consis | 0.114 | 0.057 | 0.088 | 0.096 | 0.112 | 0.136 | 0.151 | 0.088 | 0.111 | 0.133 | 0.113 | 0.094 | 0.096 | 0.104 | 0.125 | 0.111 | 0.090 | 0.073 | 0.108 | 0.079 | 0.123 | 0.105 | 0.080 | 0.076 |
| VAU-Cls | 0.092 | 0.093 | 0.111 | 0.062 | 0.104 | 0.129 | 0.121 | 0.087 | 0.097 | 0.139 | 0.075 | 0.048 | 0.082 | 0.079 | 0.117 | 0.061 | 0.067 | 0.100 | 0.081 | 0.069 | 0.133 | 0.088 | 0.049 | 0.082 |
| VAU-Act | 0.113 | 0.100 | 0.122 | 0.088 | 0.110 | 0.154 | 0.179 | 0.087 | 0.092 | 0.111 | 0.075 | 0.070 | 0.068 | 0.083 | 0.072 | 0.074 | 0.062 | 0.095 | 0.064 | 0.071 | 0.100 | 0.098 | 0.057 | 0.091 |
| VAU-Flu | 0.060 | 0.043 | 0.061 | 0.045 | 0.042 | 0.050 | 0.058 | 0.074 | 0.056 | 0.044 | 0.075 | 0.044 | 0.048 | 0.065 | 0.022 | 0.069 | 0.029 | 0.076 | 0.069 | 0.043 | 0.067 | 0.065 | 0.057 | 0.055 |
| VAU-Info | 0.117 | 0.093 | 0.139 | 0.071 | 0.085 | 0.125 | 0.154 | 0.117 | 0.094 | 0.106 | 0.092 | 0.078 | 0.057 | 0.088 | 0.061 | 0.067 | 0.071 | 0.121 | 0.081 | 0.069 | 0.111 | 0.106 | 0.103 | 0.077 |
| VAU-Fact | 0.117 | 0.093 | 0.139 | 0.071 | 0.085 | 0.125 | 0.154 | 0.117 | 0.094 | 0.106 | 0.092 | 0.078 | 0.057 | 0.088 | 0.061 | 0.067 | 0.071 | 0.121 | 0.081 | 0.069 | 0.111 | 0.106 | 0.103 | 0.077 |
| CG-CoE$_P$ | 0.087 | 0.017 | 0.139 | 0.040 | 0.067 | 0.076 | 0.000 | 0.116 | 0.000 | 0.097 | 0.104 | 0.022 | 0.067 | 0.014 | 0.176 | 0.060 | 0.045 | 0.061 | 0.155 | 0.041 | 0.056 | 0.094 | 0.014 | 0.051 |
| CG-CoE$_R$ | 0.076 | 0.000 | 0.167 | 0.018 | 0.040 | 0.042 | 0.000 | 0.043 | 0.000 | 0.120 | 0.104 | 0.003 | 0.042 | 0.021 | 0.144 | 0.034 | 0.016 | 0.000 | 0.204 | 0.037 | 0.079 | 0.111 | 0.043 | 0.051 |

Inputs (given):
**LABEL** (background only), **PRED** (to rewrite),
**ANOMALY_ANCHOR** (only truth for anomalies),
**DIMENSIONS_USED + DIMENSION_INSTRUCTIONS** (only valid edit constraints).
Goal: rewrite PRED → VARIANT such that:
Anomaly invariance (highest priority).

If ANOMALY_ANCHOR.has_anomaly = false: VARIANT must be clearly normal; no incident/accident/crime/hazard/injury/threat/violence/abnormality cues.
If has_anomaly = true: VARIANT contains all and only anchor anomalies, preserving (i) participants, (ii) triggers/conditions, (iii) event count (no merge/split), (iv) relative order, (v) numbers/times/magnitudes, (vi) causal outcomes.

Selective edits only.
**Apply only the transformations specified by DIMENSION_INSTRUCTIONS for DIMENSIONS_USED.** All other dimensions are ignored.

Conflict rule.
If a dimension edit risks changing anomalies, apply it only to non-anomalous context or use the safest approximation; anomaly invariance must hold.

Style: clear English; numbers/times may be reformatted but not changed.

Output: return **only VARIANT text (no JSON, no explanation).**

Internal check (not output): anomaly set == anchor exactly; only DIMENSIONS_USED applied.

*Figure 14.* **Prompt for CVP variant generation.**

CG-CoE (Compact Prompt Framework)

Role: You are CG-CoE, a VAU event-level evaluator.

Procedure:
1) Extract minimal anomalous events + evidence-anchored rewrites
2) Match via Strict → Flexible (class-guided boundary; supports compositional support)
3) Score event-level Precision / Recall

Input (JSON):
{"label_text":"{LABEL_TEXT}","pred_text":"{PRED_TEXT}"}

GLOBAL RULES
- Evidence-only: use only explicit text support; no invention. Missing = unknown (not contradiction).
- Minimal event: one event = one core abnormal act OR one abnormal outcome; split if multiple.
- De-dup: merge only if clearly same core anomaly (avoid paraphrase duplicates).
- Each event must include:
  * evidence_span: contiguous verbatim quote from SAME text, <=25 words
  * rewrite: conservative restatement strictly supported by evidence_span (no new entities/tools/locations/causes/numbers/outcomes)
- Do NOT extract: normal background; meta claims ("has/no anomaly", "class is ..."); explicitly normal/staged/allowed content.
- Matching is by core abnormal act/outcome (not keywords). class_hint is NOT evidence; only bounds Stage B.
- Contradiction only if explicit conflict/negation/mutually exclusive facts.

Output requirement: JSON ONLY (no extra text, no markdown).

STEP 1) Extract events
Produce two de-duplicated sets:

A) label_events (E^l) (each must include class_hint):
{"id":"L1","class_hint":"...","rewrite":"...","evidence_span":"..."}
class_hint: derive only from label_text; prefer explicit class phrase; else coarse (e.g., violence/theft/traffic accident/medical emergency). Not evidence.

Special case: if label_text clearly states no anomaly and provides no concrete abnormal incident -> label_events = []

B) pred_events (E^â) (no class_hint):
{"id":"P1","rewrite":"...","evidence_span":"..."}

STEP 2) Build support relation M ⊆ E^â × E^l
An edge means pred event supports label event (evidence-backed).
Allow compositional support: multiple pred events can cover one label event.
A pred event may cover multiple label events ONLY if its evidence supports EACH.
Match record (grouping allowed):
{"pred_ids":["P1","P2"],"label_ids":["L1","L3"],"match_stage":0,"why":"one short sentence on explicit core overlap"}
Stage A (Strict, match_stage=0):
- For all label events first.
- Match if evidence spans support the SAME core abnormal act/outcome and no explicit contradiction.
- Do NOT use class_hint for Stage A.
- Mark label events appearing in any matches[*].label_ids as covered.
Stage B (Flexible-bounded, match_stage=1):
- Only for uncovered label events L_j with class_hint h_j.
Candidate gating (ALL required):
  1) Concrete support for same core abnormal act/outcome (possibly partial under split/merge)
  2) No explicit contradiction in core act/outcome or essential roles (when stated)
  3) Within-class: consistent with h_j (no cross-class substitution)
Decision:
- If one gated P_i is sufficient -> match (P_i, L_j)
- Else choose a minimal, non-redundant set S of gated candidates; each contributes a distinct necessary anomaly fact
Forbidden weak evidence: generic "people/crowd/police appear" is never sufficient.
why: ONE short sentence describing explicit shared core act/outcome (no hypotheticals).
STEP 3) Precision / Recall
- pred event is matched if its id appears in any matches[*].pred_ids
- label event is covered if its id appears in any matches[*].label_ids
Edge cases:
1) label_events empty AND pred_events empty -> final_PR={"P":1.0,"R":1.0}, matches=[]
2) exactly one side empty -> final_PR={"P":0.0,"R":0.0}, matches=[]
3) otherwise:
  P = (# matched pred events) / (total pred events)
  R = (# covered label events) / (total label events)
Counting: each pred event counts at most once for P.
OUTPUT (JSON only):
{
  "label_events":[{"id":"L1","class_hint":"...","rewrite":"...","evidence_span":"..."}],
  "pred_events":[{"id":"P1","rewrite":"...","evidence_span":"..."}],
  "matches":[{"pred_ids":["P1"],"label_ids":["L1"],"match_stage":0,"why":"..."}],
  "final_PR":{"P":0.0,"R":0.0}
}

*Figure 15.* **Prompt used by CG-CoE.**

