# OpenReview forum: "Towards Trustworthy Video Anomaly Understanding: A Class-Guided Chain-of-Evaluation Metric and An Anomaly-focused Meta-Benchmark"
_ICML.cc/2026/Conference — ICML 2026 regular_

### Official Review · Reviewer_SQs2 · 2026-02-28

**Soundness:** 4
**Presentation:** 2
**Significance:** 3
**Originality:** 3
**Overall Recommendation:** 4
**Confidence:** 4

**Summary:**

This paper introduces CG-CoE, a class-guided chain-of-evaluation metric for Video Anomaly Understanding (VAU), together with an anomaly-focused meta-evaluation benchmark consisting of AEA (validity via human event-level Precision/Recall) and CVP (robustness under controlled stylistic perturbations). CG-CoE performs LLM-based anomalous event extraction and class-guided matching to compute event-level Precision and Recall. Extensive experiments demonstrate that CG-CoE achieves higher agreement with human annotations and improved robustness compared to existing N-gram, embedding-based, and LLM-based metrics.

**Compliance With Llm Reviewing Policy:**

Affirmed.

**Final Justification:**

This paper proposes CG-CoE and an anomaly-focused meta-evaluation framework (AEA, CVP) for VAU. I find the problem well-motivated and the proposed evaluation design sound, with consistent empirical improvements over prior metrics.

The work is meaningful in terms of originality and significance, particularly in introducing event-level, anomaly-focused evaluation. While there are minor presentation issues, they are not critical and can be easily addressed.

My initial concerns (scalability of AEA, generalization beyond Holmes-VAU, and dependence on LLM backbones) were satisfactorily addressed in the rebuttal. In particular, clarifying AEA as a protocol, providing annotation cost estimates, and showing preliminary results on CUVA improved my confidence in the approach.

Overall, the rebuttal resolved my main concerns and strengthened my evaluation. I therefore support acceptance.

**Key Questions For Authors:**

[Q1] AEA relies on detailed human event-level annotation. How do the authors plan to extend or maintain this benchmark as new anomaly domains or datasets emerge?

[Q2] The meta-evaluation is mainly conducted on Holmes-VAU. Do the conclusions about CG-CoE’s validity and robustness hold across other VAU-style datasets with different anomaly categories or narrative styles such as CUVA[1]?

[Q3] Given the rapid development of video-language and large language models, how stable are the evaluation results when using different backbone models? Would the relative ranking of metrics remain consistent?

[1]Du, Hang, et al. "Uncovering what why and how: A comprehensive benchmark for causation understanding of video anomaly." Proceedings of the IEEE/CVF Conference on Computer Vision and Pattern Recognition. 2024.

**Limitations:**

The paper does not substantially discuss limitations or broader societal considerations. While the work focuses on evaluation methodology for Video Anomaly Understanding and does not present obvious direct societal risks, a brief discussion of limitations would improve transparency. In particular, the authors could clarify:

- The dependence on LLM backbones and potential variability across model versions.

- Practical constraints in scaling or maintaining the AEA benchmark.

**Strengths And Weaknesses:**

**Strengths**

1. Clear and well-motivated problem formulation
: The paper identifies a concrete limitation of existing VAU evaluation metrics, namely their sensitivity to stylistic variation and dominance by normal content. The motivation for anomaly-focused evaluation is clearly articulated.

2. Meta-evaluation benchmark construction
: The introduction of AEA and CVP is a meaningful contribution to the VAU community. The event-level human annotation protocol is carefully designed, and the robustness evaluation via controlled variant pairs is systematic.

3. Coherent metric design
: CG-CoE follows a clear pipeline (minimal event extraction, strict-then-flexible matching, class-guided tolerance boundary) and is supported by ablation studies that justify each component.

4. Empirical improvements
: On the proposed meta-benchmark, CG-CoE consistently outperforms prior metrics in terms of both validity (correlation with human event-level Precision/Recall) and robustness to stylistic perturbations.

**Weaknesses**
1. Scalability of meta-benchmark construction
: Although CG-CoE itself is fully automated, the AEA benchmark requires substantial human annotation effort. It remains unclear how easily the meta-evaluation framework can be extended to larger or more diverse datasets.

2. Presentation issues
: The manuscript contains minor but noticeable issues (e.g., typos in figures [Trunk-> Truck] and potentially confusing reuse of notation such as $E_{l},E_{\hat{a}}$ for both AEA and CG-CoE) which reduce clarity. The exposition could be streamlined for better readability.

---

> ### Author Rebuttal · Authors · 2026-03-31
>
> > **Q1: Scalability of meta-benchmark construction**.
>
> We thank the reviewer for this important question. We believe AEA is reasonably scalable and can be extended to most VAU datasets, because many existing VAU benchmarks already provide anomaly-related descriptions that are suitable for event-level annotation. Since such anomaly descriptions are central to VAU itself, future datasets are also likely to remain compatible with this design.
>
> Extending AEA does not require new data collection. It is constructed from existing VAU samples and model outputs that are already available in standard experiments. The main additional cost is human annotation rather than data creation. In our pilot study, about 30 minutes of training was sufficient, and the total annotation cost was about 4 hours for 2 annotators, corresponding to roughly 40 videos per hour.
>
> **Therefore, although AEA is not annotation-free, it has lightweight data requirements and a manageable human cost, making it reasonably scalable in practice.**
>
> > **Q2: Clarity and presentation**.
>
> We thank the reviewer for pointing out these issues. We agree that, although they are relatively minor, they can still affect the clarity and readability of the manuscript. In the revision, we will correct the typographical errors, avoid confusing notation reuse, and further improve the overall presentation.
>
> > **Q3: Specific extension and maintenance strategy**.
>
> We thank the reviewer for this important question. AEA only requires anomaly-centered event descriptions for human event-level annotation, and such information is already available in many existing VAU datasets. For example, multiple VAU datasets already provide suitable anomaly-related descriptions, including CUVA, ECVA, HAWK, VAD-R1, and VAU-R1.
>
> Extending AEA to a new dataset mainly involves three steps: (1) using the existing anomaly-related descriptions as references, (2) collecting VAU model outputs on the same samples, and (3) applying our human event-level annotation protocol. Therefore, AEA is not limited to a specific benchmark, but can be naturally extended to new datasets and anomaly domains.
>
> **Overall, AEA is a protocol rather than a fixed benchmark, and can be applied to new VAU datasets as they emerge.**
>
> > **Q4: Generalization across different VAU datasets**.
>
> We thank the reviewer for this question. **CG-CoE is not limited to Holmes-VAU and is designed to generalize to VAU-style datasets such as CUVA.**
>
> This generalization comes from two aspects. First, CG-CoE does not assume a closed set of anomaly categories, but uses anomaly labels or descriptions from the dataset as `class_hint`, allowing it to adapt to new anomaly types. Second, by focusing on anomalous event matching rather, it is less sensitive to dataset-specific wording and annotation style. To provide preliminary evidence, we applied our AEA annotation protocol to CUVA and built a small validation set with 200 samples. Since this evidence comes from only one external dataset with limited size, it should be viewed as preliminary rather than conclusive.
>
> | Validation Set     | CG-CoE | HAWK | VAD-R1 |
> | ------------------ | ------ | ---- | ------ |
> | CUVA (200 samples) | 0.91   | 0.81 | 0.87   |
>
>
>
> > **Q5: The ablations**.
>
> We thank the reviewer for this important question. Due to space limitations, we refer the reviewer to our response to Reviewer `77jw`, `Q2` for the full ablation results.
>
> Briefly, we agree that both the backbone LLM and the prompt design can affect absolute metric values. However, our ablations show that the overall conclusion remains stable: CG-CoE consistently outperforms competing metrics across different backbones and prompt variants. **This suggests that its advantage mainly comes from the proposed evaluation design, rather than reliance on a specific LLM or fragile prompt tuning.**
>
> > **Q6: Limitations**
>
> Our current metric assumes that the model output contains sufficient anomalous-event information for semantic evaluation. Under this setting, the evaluator can assess whether anomalous events are correctly understood by comparing the prediction with the reference at the event level. However, when the answer is too incomplete or underspecified to support reliable judgment, this text-only setting becomes inherently limited. In such cases, accurate evaluation may require direct access to the video content rather than relying solely on the textual response.
>
> In addition, the current benchmark is mainly constructed based on Holmes-VAU. Although this setting is sufficient to support our initial meta-evaluation study, broader validation across more VAU datasets with different anomaly categories, narrative styles, and annotation protocols would further strengthen the generality of our conclusions.
>
> We therefore view extending the evaluator to use video evidence and expanding the benchmark to more VAU datasets as important future work.

---

> > ### Author Rebuttal · Reviewer_SQs2 · 2026-04-03
> >
> > Thank you for the clear and constructive rebuttal.
> > The responses sufficiently addressed my concerns regarding scalability, generalization, and backbone dependency.
> >
> > In particular, the clarification that AEA is a protocol rather than a fixed benchmark, along with the preliminary validation on CUVA and discussion on annotation cost, improved my understanding of the practicality and extensibility of the framework.
> >
> > I will update my score accordingly.

---

> > > ### Author Response · Authors · 2026-04-04
> > >
> > > We sincerely thank the reviewer for recognizing that our rebuttal has adequately addressed the concerns on scalability, generalization, and backbone dependency. We especially appreciate the reviewer’s acknowledgment that clarifying AEA as a protocol rather than a fixed benchmark, together with the preliminary validation on CUVA and the discussion of annotation cost, improves the understanding of the framework’s practicality and extensibility. We are grateful for this positive reassessment and for the updated score.

---

### Official Review · Reviewer_SsBG · 2026-03-06

**Soundness:** 4
**Presentation:** 3
**Significance:** 3
**Originality:** 3
**Overall Recommendation:** 4
**Confidence:** 4

**Summary:**

A Class-Guided Chain-of-Evaluation metric is proposed to decouple anomaly semantics from expression styles. And an anomaly-focused meta-evaluation benchmark is also introduced for video anomaly understanding. Extensive experiments show that CG-CoE achieves SOTA validity and robustness, significantly outperforming existing N-gram-based, embedding-based, and LLM-based metrics.

**Compliance With Llm Reviewing Policy:**

Affirmed.

**Key Questions For Authors:**

Please see the Weaknesses.

**Limitations:**

The authors don’t provide a detailed discussion of the limitations and potential negative societal impact of this work in the manuscript.

**Strengths And Weaknesses:**

Strengths:
1. This paper correctly identifies that the current VAU metrics are fundamentally flawed: they are dominated by normal content and sensitive to expression style. This insight is crucial for the field, as using inappropriate metrics can lead to incorrect conclusions about model performance.
2. A VAU-specific metric CG-CoE is introduced to evaluate anomaly understanding by extracting anomalous events and performing evidence-backed matching under a class-guided tolerance boundary.
3. The first anomaly-focused meta-evaluation benchmark is constructed for trustworthy VAU evaluation and model deployment. This establishes a new standard for evaluation in the field and will likely drive further progress.

Weaknesses:
1. The metric and benchmark are designed to evaluate only the textual descriptions generated by VAU models. Designing mechanisms to evaluate a model's temporal or spatial localization capabilities would further enhance the reliability of its reasoning content.
2. The class-guided matching in CG-CoE and the annotation process in AEA rely on a predefined set of anomaly classes. In a truly open-world setting, anomalies may not fit into these predefined categories.
3. The manuscript lacks a description of the limitations of the CG-CoE and anomaly-focused meta-evaluation benchmark.

---

> ### Author Rebuttal · Authors · 2026-03-31
>
> > **Q1: The extension to temporal and spatial localization**.
>
> We thank the reviewer for this valuable suggestion. We fully agree that temporal and spatial localization are important components of Video Anomaly Understanding (VAU). In fact, prior VAU benchmarks have already provided relatively mature protocols for these capabilities. For example, some datasets evaluate temporal localization using anomaly intervals and IoU-style measures, while others provide spatial grounding annotations to assess spatial localization performance. We agree that incorporating temporal or spatial localization analysis could further enrich the assessment of VAU models.
>
> At the same time, we would like to clarify that localization metrics alone do not fully capture whether a model truly understands anomalous events at the semantic level. A model may correctly identify when or where an anomaly occurs, yet still fail to accurately explain what the anomaly is. Our work therefore focuses on this complementary but still underexplored problem: how to reliably evaluate anomaly understanding from the textual outputs of VAU models.
>
> This complementary problem is itself nontrivial, because existing evaluators may suffer from several systematic issues when comparing predictions with references. They can be distracted by lengthy normal-context descriptions and therefore underweight anomalous content; they may rely on coarse overall similarity or shallow matching that confuses surface overlap with correct anomaly understanding; and prior work has also shown that LLM-based evaluation can exhibit stylistic preference and related biases.
>
> Motivated by these challenges, our benchmark and metric are designed to provide a more reliable textual evaluation of anomaly understanding. Thus, our goal is not to replace temporal or spatial localization evaluation, but to complement existing localization protocols with a benchmark and metric that focus specifically on anomaly semantics. We believe this complementary perspective is important for advancing VAU beyond identifying when or where anomalies occur, toward understanding what the anomaly actually is.
>
>
> > **Q2: Adaptation to open-world anomalies**.
>
> We thank the reviewer for raising this important concern. We agree that handling open-world anomalies is important for evaluating whether a VAU benchmark or metric can remain useful in realistic scenarios.
>
> We would like to clarify that neither AEA nor CG-CoE is limited to a fixed closed set of anomaly classes. In benchmark construction, the anomaly type is taken from the label information provided by the underlying VAU dataset, while the final judgment is made through human annotation and event-level matching. Therefore, when extending the benchmark to a new dataset, annotators can naturally incorporate newly introduced anomaly categories rather than being restricted to a predefined taxonomy.
>
> Similarly, CG-CoE does not assume a fixed class inventory. Instead, it uses the anomaly type in the reference label as a `class_hint` to guide evaluation. When a new anomaly type appears, the metric is not forced to map it into an existing class set. Rather, the new type can be extracted from the label information and used in the same evaluation procedure. **Therefore, our current claim is better described as extensibility to new anomaly categories under dataset-supported settings, rather than a complete solution to unrestricted open-world anomaly evaluation.**
>
>
> > **Q3: The limitations**.
>
> Our current metric assumes that the model output contains sufficient anomalous-event information for semantic evaluation. Under this setting, the evaluator can assess whether anomalous events are correctly understood by comparing the prediction with the reference at the event level. However, when the answer is too incomplete or underspecified to support reliable judgment, this text-only setting becomes inherently limited. In such cases, accurate evaluation may require direct access to the video content rather than relying solely on the textual response.
>
> In addition, the current benchmark is mainly constructed based on Holmes-VAU. Although this setting is sufficient to support our initial meta-evaluation study, broader validation across more VAU datasets with different anomaly categories, narrative styles, and annotation protocols would further strengthen the generality of our conclusions.
>
> We therefore view extending the evaluator to use video evidence and expanding the benchmark to more VAU datasets as important future work.

---

> > ### Author Rebuttal · Reviewer_SsBG · 2026-04-07
> >
> > Thank you for the author's response, but my concerns regarding the validity evaluation of the textual descriptions still remain unresolved.

---

> > > ### Author Response · Authors · 2026-04-07
> > >
> > > We thank the reviewer for the further clarification. We agree that evaluating textual descriptions alone does not fully capture complete VAU ability, and cannot replace temporal or spatial localization evaluation. Our claim is narrower: the proposed benchmark evaluates ability from textual descriptions, which is an important but partial dimension of VAU.
> > >
> > > This scope is also consistent with recent VAU literature. For example, VAU-R1 explicitly decomposes VAU into complementary tasks including temporal grounding, anomaly reasoning, and anomaly classification, with each task evaluated by its own metric. In other words, anomaly reasoning from textual outputs is treated there as a legitimate VAU dimension, rather than a purely stylistic byproduct.
> > >
> > > Moreover, prior results suggest that text-level anomaly reasoning is not disconnected from temporal grounding. In VAU-R1, improvements in reasoning quality are accompanied by improvements in temporal localization in several settings, especially on the out-of-distribution UCF-Crime benchmark. For the same backbone (Qwen2.5-VL-3B), moving from SFT to RFT increases the VAU-Eval total from **10.89** to **25.49** on UCF-Crime, while mIoU simultaneously rises from **4.98** to **16.80** under *w/o think* inference and from **5.76** to **9.21** under *w/ think* inference. A similar trend is also observed on MSAD, where the VAU-Eval total increases from **15.96** to **33.38**, while mIoU under *w/o think* rises from **30.65** to **35.77**.
> > >
> > > ### Evidence from VAU-R1
> > >
> > >
> > >
> > > | Dataset   | Model Variant       | VAU-Eval Total ↑ | mIoU (w/o think) ↑ | mIoU (w/ think) ↑ |
> > > | --------- | ------------------- | ---------------: | -----------------: | ----------------: |
> > > | MSAD      | Qwen2.5-VL-3B       |            32.47 |              21.27 |             13.00 |
> > > | MSAD      | Qwen2.5-VL-3B + SFT |            15.96 |              30.65 |             35.17 |
> > > | MSAD      | Qwen2.5-VL-3B + RFT |            33.38 |              35.77 |             30.70 |
> > > | UCF-Crime | Qwen2.5-VL-3B       |            25.10 |              10.91 |              7.68 |
> > > | UCF-Crime | Qwen2.5-VL-3B + SFT |            10.89 |               4.98 |              5.76 |
> > > | UCF-Crime | Qwen2.5-VL-3B + RFT |            25.49 |              16.80 |              9.21 |
> > >
> > >
> > >
> > > Table values above are taken from VAU-R1 Table 1 and Table 2.
> > >
> > > Similar evidence also appears in Vad-R1: stronger training strategies that improve anomaly reasoning metrics also improve anomaly detection metrics. For instance, from the base Qwen2.5-VL model to `+SFT+AVA-GRPO`, ROUGE-L improves from **0.477** to **0.501**, precision from **0.768** to **0.882**, and mIoU from **0.567** to **0.713**.
> > >
> > > ### Evidence from VAD-R1
> > >
> > > | Training Strategy | ROUGE-L ↑ | Precision ↑ | mIoU ↑ | R@0.5 ↑ |
> > > | ----------------- | --------: | ----------: | -----: | ------: |
> > > | Qwen2.5-VL        |     0.477 |       0.768 |  0.567 |   0.563 |
> > > | +SFT              |     0.429 |       0.712 |  0.612 |   0.599 |
> > > | +AVA-GRPO         |     0.486 |       0.810 |  0.675 |   0.661 |
> > > | +SFT+AVA-GRPO     |     0.501 |       0.882 |  0.713 |   0.706 |
> > >
> > >
> > >
> > > These results do not imply that textual evaluation fully substitutes for temporal/spatial localization. Rather, they provide convergent evidence that better textual anomaly reasoning is meaningfully associated with stronger temporal grounding/detection. **Therefore, we believe that evaluating textual descriptions is justified as a valid and practically important way to assess one dimension of VAU , although not a complete proxy for end-to-end VAU ability.** We will revise the manuscript to make this scope and limitation explicit.
> > >
> > >
> > >
> > > We hope that these additional analyses help alleviate the reviewer’s concern regarding validity and support a more favorable reassessment of our paper. If there are still remaining concerns, we would be happy to continue clarifying them in a constructive manner.

---

### Official Review · Reviewer_1mru · 2026-03-11

**Soundness:** 3
**Presentation:** 2
**Significance:** 3
**Originality:** 3
**Overall Recommendation:** 4
**Confidence:** 3

**Summary:**

The paper presents a new metric to evaluate the video anomaly understanding, together with a two meta evaluation benchmarks. The metric (CG-CoE) extracts anomaly events based on LLMs and matches them to a class-specific semantic tolerance boundary.

**Compliance With Llm Reviewing Policy:**

Affirmed.

**Final Justification:**

given the new results posted in rebuttal, with a different llm family showing that the method is not entirely dependent on the llm and provides resonable  scores with other llms too, i increased my score.

**Key Questions For Authors:**

pleas answer weaknesses

**Limitations:**

not at all. limitations must be discussed.

**Strengths And Weaknesses:**

Strengths:
- The paper focuses on an important problem, since lack of consistent evaluation, or metrics that do not actually correlates with human-judgment hurt the development of the algorithms.
- The paper provides two datasets manually annotated.
- Both metrics highly align with human-judgment and it is demonstrated that the metric accounts for anomalies not for expression styles (with dataset CPV)


Weaknesses:
- The CG-CoE overly relies on LLMs, and I have not seen an ablation based on the LLM used to compute the metric.
- The method section is hard to follow, the intent behind the equation are not explain.
- The limitation of the metric is not discussed. Since it relies on LLM to extract anomaly events and classes how the LLM performance influence the metric? Should anyone just use deepseek?
- Since they constructed two video anomaly understanding data set, visual examples from the data sets are missing along details about how they were constructed


Overall: The intro was nicely written and easy to follow, however when moving to Meta-Evaluation Benchmark, many steps without the intuition behind it made it hard to understand. I do acknowledge the importance of the metric and the data sets, but it is hard for me to decide if this paper mitigates these issues properly.

---

> ### Author Rebuttal · Authors · 2026-03-31
>
> > **Q1:The ablation on LLM and prompt**.
>
> We thank the reviewer for this important question. Due to space limitations, we refer the reviewer to our response to Reviewer `77jw`, `Q2`, for the full ablation tables.
>
> Briefly, our experiments show that CG-CoE is influenced by the choice of judge LLM and prompt, but this influence mainly appears in the **absolute score values**, rather than in the **relative conclusions**. Across different judge backbones, CG-CoE consistently outperforms the competing metrics. This suggests that its advantage mainly comes from the **evaluation design itself**, rather than from reliance on any specific LLM.
>
> We also conduct systematic prompt ablations using semantically equivalent prompt variants. Although prompt changes cause some fluctuation in absolute scores, CG-CoE remains consistently strong across variants. This indicates that its effectiveness does not come from fragile prompt engineering, but from the anomaly-centered evaluation pipeline.
>
>
>
> > **Q2: The rationale of the method**.
>
> We thank the reviewer for this helpful comment. CG-CoE is motivated by a mismatch between prior metrics and the goal of VAU. In VAU, the key question is whether anomalous events are correctly understood. However, many existing metrics can be distracted by anomaly-irrelevant content, so scores may reflect similarity rather than anomaly alignment. They may also fail to match descriptions of the same anomaly when different aspects are emphasized.
>
> CG-CoE addresses this by shifting evaluation from whole-answer similarity to anomaly-centered event matching. Anomalous event extraction suppresses irrelevant normal content. Class-guided matching introduces a class-dependent tolerance boundary, so equivalent descriptions can still align despite differences in wording or granularity. Event-level Precision and Recall separate anomaly correctness from anomaly coverage, making the score more interpretable.
>
>
> > **Q3: The limitations and practical use**.
>
> We thank the reviewer for this important question. We agree that the limitations of CG-CoE should be discussed more explicitly.
>
> First, CG-CoE is still a text-only evaluator. It assumes that the model output contains sufficient anomalous-event information for semantic judgment. When the prediction is too incomplete or underspecified, or when the anomaly can only be reliably disambiguated from the video itself, text-only evaluation becomes limited. Extending CG-CoE toward multimodal evaluation is therefore an important future direction.
>
> Second, CG-CoE is not fully invariant to the judge LLM. As shown in Q1, changing the backbone affects the absolute scores, but not the main conclusion: CG-CoE consistently outperforms prior metrics across the tested models. In practice, DeepSeek-V3 can be used as a practical default in our setup. However, CG-CoE is not tied to DeepSeek specifically: other strong LLMs can also be used, and the main conclusions remain consistent across them.
>
> > **Q4: The visual examples and construction transparency**.
>
> We thank the reviewer for this suggestion.We first clarify that our benchmark consists of two subsets, AEA and CVP, rather than two separate datasets. AEA evaluates metric validity through human event-level Precision/Recall annotations, while CVP tests robustness under anomaly-preserving stylistic perturbations.
>
> The manuscript includes the core workflow. AEA is built from Holmes-VAU test samples, where two annotators independently extract and match anomalous events to derive human event-level Precision/Recall, with disagreements resolved through discussion. CVP is built from the same source pool by extracting anomaly anchors, generating controlled variants, and applying automatic verification and manual checking.
>
> AEA contains 865 final prediction-label pairs, derived from 900 pairs over 150 Holmes-VAU videos after filtering irrelevant outputs. Its pre-adjudication agreement reaches ICC 0.85/0.82 for Precision/Recall, indicating reliable annotation. CVP contains 898 valid groups after anomaly-anchor-based rewriting, verification, retry, and filtering.
>
> We agree, however, that these details are not yet sufficiently inspectable. To address this, we will revise the paper by adding more visualized cases and a clearer benchmark overview, including: (1) representative AEA examples with video key frames, labels, predictions, extracted anomalous events, event matching, and final human Precision/Recall; (2) representative CVP examples with original answers, anomaly anchors, selected style dimensions, rewritten variants, and verification outcomes; and (3) an end-to-end construction flow summarizing data source, annotation/generation, filtering, and quality-control steps. For transparency, all of these materials have been organized on an anonymous supplementary page: https://anonymous.4open.science/r/rebuttal-D5FB

---

> > ### Author Rebuttal · Reviewer_1mru · 2026-04-04
> >
> > Thank the authors for their efforts.
> > I thank the authors for their answers.  However, my concerns about the limitations of the methods and the reliance on LLM is not fully covered.

---

> > > ### Author Response · Authors · 2026-04-05
> > >
> > > We thank the reviewer for further clarifying this concern. We understand that the central issue is not simply whether CG-CoE remains stable across several strong LLM backbones, but how the capability of the judge LLM itself influences the metric.
> > >
> > > To directly probe this question, we conduct an additional pilot study using different model scales within the same LLM family (Qwen3.5), evaluated on 100 samples:
> > >
> > > | Judge LLM    | AEA ρ↑ | CVP 𝔼[Δ]↓ |
> > > | ------------ | -----: | --------: |
> > > | Qwen3.5-0.8B |    N/A |       N/A |
> > > | Qwen3.5-2B   |    N/A |       N/A |
> > > | Qwen3.5-4B   |    N/A |       N/A |
> > > | Qwen3.5-9B   |  0.861 |     0.121 |
> > > | Qwen3.5-27B  |  0.858 |     0.110 |
> > > | Qwen3.5-35B  |  0.871 |     0.083 |
> > > | Qwen3.5-122B |  0.853 |     0.092 |
> > > | Qwen3.5-197B |  0.862 |     0.070 |
> > >
> > > We mark 0.8B--4B as N/A because these smaller models frequently produced severely corrupted or otherwise uninterpretable outputs, making their evaluations unreliable. **This suggests that CG-CoE does depend on the judge LLM’s instruction-following and structured semantic reasoning ability.**
> > >
> > > At the same time, the results indicate a threshold effect rather than a monotonic scaling trend. Once the judge reaches a sufficient capability level (here, around 9B and above), the intermediate reasoning steps become much more stable, and CG-CoE operates reliably across different backbones within this regime. **In other words, CG-CoE is not tied to any single model such as DeepSeek. Rather, it requires an LLM that can reliably follow instructions and perform structured semantic reasoning.**
> > >
> > > We hope that these additional analyses help alleviate the reviewer’s concern about the effect of different LLM backbones and support a more favorable reassessment of our paper. If any concerns remain, we would be happy to continue addressing them in a constructive manner.

---

### Official Review · Reviewer_77jw · 2026-03-12

**Soundness:** 2
**Presentation:** 2
**Significance:** 2
**Originality:** 2
**Overall Recommendation:** 3
**Confidence:** 2

**Summary:**

This paper proposes CG-CoE, a class-guided chain-of-evaluation metric designed for Video Anomaly Understanding (VAU). The method extracts anomalous events from predictions and labels, aligns them under class-specific tolerance boundaries, and computes event-level precision and recall to evaluate anomaly understanding. In addition, the authors introduce an anomaly-focused meta-evaluation benchmark consisting of two subsets: AEA for measuring validity through human event-level annotations and CVP for evaluating robustness to stylistic perturbations. Experimental results show that CG-CoE achieves higher correlation with human judgments compared with existing metrics.

**Compliance With Llm Reviewing Policy:**

Affirmed.

**Key Questions For Authors:**

See weaknesses

**Limitations:**

I do not see any discussion on limitations.

**Strengths And Weaknesses:**

Strengths

1. The paper considers the reliability of evaluation metrics in Video Anomaly Understanding, which is an important but relatively underexplored problem.

2. The proposed AEA and CVP datasets attempt to measure metric validity and robustness through human annotations and controlled stylistic variants.

Weaknesses

1. Although CG-CoE introduces a class-guided matching strategy, the overall framework largely follows the common extract–match–score paradigm used in many structured evaluation metrics. The main difference lies in the addition of class-based tolerance boundaries and event-level matching, which appear to be incremental modifications.

2. The metric pipeline depends on large language models to extract anomalous events, rewrite them into a uniform style, and perform class-guided matching. This raises concerns regarding reproducibility, stability, and sensitivity to prompt design. Small changes in the LLM backbone or prompts may significantly influence the evaluation results, yet this issue is not sufficiently discussed.

3. The proposed meta-evaluation benchmark relies on human annotations derived from outputs of several VAU models and additional synthetic variants generated using language models. Because the benchmark is partially constructed from model outputs and LLM-based rewriting, there is a risk that the evaluation setup implicitly favors metrics designed with similar assumptions.

---

> ### Author Rebuttal · Authors · 2026-03-31
>
> > **Q1: The novelty of our methodology**
>
> We thank the reviewer for this comment. We clarify that our contribution is not merely a modification of an existing metric pipeline, but, to the best of our knowledge, the first anomaly-focused meta-evaluation benchmark for Video Anomaly Understanding (VAU). Both the benchmark and CG-CoE target the core need of VAU evaluation: focusing on anomalous events.
>
> In contrast, existing metrics are often only asked to focus on anomaly, without an explicit mechanism. They can still be distracted by anomaly-irrelevant content, so scores may reflect similarity rather than anomaly alignment (a quantitative example is provided in https://anonymous.4open.science/r/rebuttal-D5FB). They also often lack anomaly-specific semantic understanding, and thus fail to match descriptions of the same anomalous event when different aspects are emphasized.
>
> To address this, the event-level representation makes evaluation explicit by extracting minimal anomalous events and reducing irrelevant interference. The class-guided tolerance boundary introduces prior knowledge, enabling the metric to recognize diverse yet valid descriptions within the same anomaly class. **Therefore, CG-CoE is not a simple incremental change to the extract–match–score framework, but a shift from vague scoring to explicit, interpretable, and task-specific evaluation for VAU.**
>
> > **Q2:The ablation and the reproducibility**
>
> We thank the reviewer for raising this important concern. We agree that LLM-based metrics, including CG-CoE, can be affected by both backbone choice and prompt wording in their absolute values.
>
> To examine this, we conduct two additional analyses. First, we re-evaluate CG-CoE and representative LLM-based baselines using multiple backbone models. While the absolute scores vary, **CG-CoE remains the top-performing metric on AEA and CVP under all tested backbones.** Second, following prior multi-prompt evaluation practice, we construct semantically equivalent prompt variants at lexical, block-preserving, and full-paraphrase levels. **Across these variants, CG-CoE shows stable relative ranking and competitive worst-case performance.**
>
> | Model             | AEA-CG-CoE ↑ | AEA-HAWK ↑ | AEA-VAD-R1 ↑ | CVP-CG-CoE ↓ | CVP-HAWK ↓ | CVP-VAD-R1 ↓ |
> | ----------------- | ------------ | ---------- | ------------ | ------------ | ---------- | ------------ |
> | Claude-Sonnet-4-6 | **0.751**    | 0.700      | 0.720        | **0.023**    | 0.044      | 0.034        |
> | DeepSeek-V3.2     | **0.766**    | 0.688      | 0.655        | **0.027**    | 0.045      | 0.050        |
> | Gemini-3.1        | **0.751**    | 0.682      | 0.719        | **0.031**    | 0.052      | 0.034        |
> | GPT-5.4           | **0.718**    | 0.685      | 0.701        | **0.035**    | 0.040      | 0.042        |
>
> | Setting      | AEA ↑  | CVP ↓  |
> | ------------ | ------ | ------ |
> | Median       | 0.7810 | 0.0274 |
> | 95% quantile | 0.7888 | 0.0310 |
> | Worst-case   | 0.7676 | 0.0317 |
>
> Regarding reproducibility, CG-CoE uses a fixed prompt template, a strictly defined JSON schema, and deterministic post-processing rules. We will clarify these implementation details and add the new backbone/prompt ablations in the revision. At the same time, we will explicitly discuss as a limitation that absolute metric values can vary with the LLM backend, even though the relative ranking is stable in our experiments.
>
>
>
> > **Q3:The fairness of benchmark and CG-CoE**
>
> We thank the reviewer for this important concern. We agree that benchmark construction should avoid implicitly favoring the proposed metric.
>
> For AEA, the target judgments are not produced by CG-CoE or by any LLM evaluator. They are human event-level Precision/Recall annotations obtained through a two-annotator extract-and-match protocol with adjudication. The model outputs are used only as candidate responses to be evaluated, while the supervisory target remains independent human judgment.
>
> For CVP, the goal is to test whether a metric is robust to anomaly-preserving stylistic variation. To avoid semantic drift, each rewrite is constrained by an anomaly anchor, verified for anomaly invariance and constraint satisfaction, and further manually checked. Thus, CVP does not encode the internal mechanism of CG-CoE; rather, it operationalizes a task-level requirement that any reliable VAU metric should satisfy, namely, that scores should depend on anomaly semantics rather than surface wording.
>
> **Therefore, the benchmark is not designed to favor CG-CoE. Rather, it operationalizes task-level requirements that any reliable VAU metric should satisfy, namely, validity with respect to human anomaly understanding and robustness to anomaly-preserving stylistic variation.**

---

> > ### Author Rebuttal · Reviewer_77jw · 2026-04-03
> >
> > Thank the authors for their efforts. However, concerns about the fairness of the benchmarking process remain. It would be beneficial to include more quantitative results to support their claim.

---

> > > ### Author Response · Authors · 2026-04-05
> > >
> > > We thank the reviewer for pointing out that the fairness concern requires quantitative evidence beyond our procedural clarification. We agree that, if the benchmark conclusions were overly dependent on a limited set of VAU source models or on a specific LLM-based rewriting pipeline, unintended bias could indeed be introduced. To address this concern directly, we conducted additional experiments along three axes.
> > >
> > > ## AEA: source-model diversity
> > >
> > > Our original AEA candidates were collected from a limited set of VAU models. We therefore expanded the candidate pool by additionally sampling outputs from three stronger and stylistically diverse vision-language models (GPT-5.4, Gemini-3.1, and Claude-Sonnet-4-6) on 20 videos, yielding 60 additional responses. These responses were annotated using the same event-level protocol. **On this expanded subset, CG-CoE still achieves the best overall agreement with human judgment, indicating that its advantage is not tied to a particular source-model distribution.**
> > >
> > > | Source Model      |  CG-CoE ↑ |    HAWK ↑ |  VAD-R1 ↑ |
> > > | ----------------- | --------: | --------: | --------: |
> > > | Claude-Sonnet-4-6 |     0.860 |     0.698 |     0.659 |
> > > | Gemini-3.1        |     0.928 |     0.896 |     0.887 |
> > > | GPT-5.4           |     0.920 |     0.836 |     0.651 |
> > > | **Average**       | **0.903** | **0.810** | **0.732** |
> > >
> > > ## AEA: prompt diversity
> > >
> > > We further tested whether the response-generation prompt itself could bias the benchmark. To this end, we used multiple anomaly-description prompts adapted from CUVA, Holmes-VAU, and HAWK, producing 120 additional responses under diverse instruction styles. These responses were again annotated using the same protocol. **CG-CoE remains consistently superior on this subset as well, suggesting that its advantage is stable under prompt variation rather than arising from a specific prompt design.**
> > >
> > > | Validation Set |  CG-CoE ↑ |    HAWK ↑ |  VAD-R1 ↑ |
> > > | -------------- | --------: | --------: | --------: |
> > > | **120 pairs**  | **0.811** | **0.719** | **0.530** |
> > >
> > > ## CVP: rewrite-backbone diversity
> > >
> > > Our original CVP rewrites were generated using a single LLM. To test whether the conclusion depends on that choice, we additionally used GPT-5.4, Gemini-3.1, Claude-Sonnet-4-6, Llama-4-Scout, and DeepSeek-V3.2 to produce anomaly-preserving rewrites on 20 examples under the same anchor-constrained rewriting protocol, followed by manual verification. **Across these rewrite backbones, CG-CoE continues to show the strongest robustness, indicating that the result is not tied to a specific rewriting model.**
> > >
> > > | Rewrite Model     |  CG-CoE ↓ |    HAWK ↓ |  VAD-R1 ↓ |
> > > | ----------------- | --------: | --------: | --------: |
> > > | Claude-Sonnet-4-6 |     0.023 |     0.044 |     0.032 |
> > > | DeepSeek-V3.2     |     0.034 |     0.066 |     0.057 |
> > > | Gemini-3.1        |     0.020 |     0.077 |     0.037 |
> > > | GPT-5.4           |     0.047 |     0.050 |     0.040 |
> > > | Llama-4-Scout     |     0.010 |     0.061 |     0.061 |
> > > | **Average**       | **0.027** | **0.059** | **0.045** |
> > >
> > > Overall, although these additional experiments do not cover all possible benchmark constructions, **they provide direct quantitative evidence that the superiority of CG-CoE remains stable under broader source-model, prompt, and rewrite-model variations, rather than arising from a particular construction choice.**
> > >
> > > We hope that these additional analyses help alleviate the reviewer’s concern regarding fairness and support a more favorable reassessment of our paper. If there are still remaining concerns, we would be happy to continue clarifying them in a constructive manner.

---

### Decision · Program_Chairs · 2026-04-30

**Decision:**

Accept (regular)

**Comment:**

The reviews were generally positive about the problem choice and the potential value of an anomaly-focused evaluation framework, with several reviewers viewing the benchmark and metric as a meaningful contribution to VAU evaluation. The main sticking point throughout the discussion was the dependence on the judge LLM and, related to that, whether the benchmark construction might bias the conclusions. The authors did a serious rebuttal effort here: they added cross-backbone and prompt ablations, additional fairness checks over source models and rewrite models, a pilot study showing that very small judge models fail while sufficiently capable models behave much more stably, and clarifications on scope, scalability, and generalization to CUVA. This resolved the concerns for some reviewers, including one who explicitly moved to support acceptance after the rebuttal, including Reviewer SQs2, while Reviewer 1mru still felt that the LLM dependence and limitations were not fully settled. I agree with the concern that the method quality still depends in part on the capability of the evaluator LLM, and I do not think the paper completely closes that issue. At the same time, the rebuttal makes a credible case that the core ranking behavior is reasonably stable once the judge model clears a capability threshold, and the paper addresses an evaluation problem that matters in this area.